mechanical engineering

sand casting, performance parameters, optimization, prediction, grey relational analysis, back propagation neural network

**Authors for correspondence:**
Kaili Xu
e-mail: xklsafety@163.com
Xiwen Yao
e-mail: yxw_20061005@126.com

# Optimization of sand casting performance parameters and missing data prediction

## Qingwei Xu, Kaili Xu, Li Li and Xiwen Yao

Key Laboratory of Ministry of Education on Safe Mining of Deep Metal Mines, School of Resources and Civil Engineering, Northeastern University, Shenyang 110819, People's Republic of China

QX, 0000-0003-1621-7995; KX, 0000-0001-5054-1241; XY, 0000-0002-2780-4143

Due to a wide range of applications, sand casting occupies an important position in modern casting practice. The main purpose of this study was to optimize the performance parameters of sand casting based on grey relational analysis and predict the missing data using back propagation (BP) neural network. First, the influence of human factors was eliminated by adopting the objective entropy weight method, which also saved manpower. The larger variation degree in the evaluation indicators, indicating that the evaluated projects had good discrimination in this regard, the larger weight should be given to these evaluation indicators. Second, the performance parameters of sand casting were optimized based on grey relational analysis, providing a reference for sand milling. The larger the grey relational degree, the closer the evaluated project was to the ideal project. Third, this paper provided a new method for determining the number of hidden neurons in a network according to the mean square error of training samples, and venting quality was predicted based on BP neural network. The relevant theory was deduced before predicting missing data, such that there will be a general understanding regarding the prediction principle of BP neural network. Fourth, to demonstrate the validity of BP neural network adopted in the process of missing data prediction, grey system theory was applied to compare the result of missing data prediction.

## 1. Introduction

The industries and national economy in China have rapidly developed [1,2]. The foundry industry is the basis of modern equipment manufacturing, which occupies a very important position in the national economy [3,4]. However, during its service in social development, the foundry industry has also caused some negative effects, such as environment pollution [5–7]

and casualties [8], which have a certain relationship with the foundry materials, and countermeasures should be adopted to eliminate the serious results [9,10]. In the foundry industry, the proportion of sand casting has increased nearly 90% [11], and it is of great significance to select appropriate performance parameters of sand casting for the safe foundry production.

For optimizing design in sand casting, the main research works focus on optimization of riser design [12,13] and process parameters [14]. When optimizing the sand casting performance parameters, usually one parameter in batch $A$ reaches the ideal value, and another parameter in batch $B$ reaches the ideal value, but not all the parameters in one batch can reach the ideal values at the same time. Therefore, it is of great significance to optimize the performance parameters in a single batch. The common optimization method of sand casting performance parameters is Taguchi's method [14], but this method has some disadvantages. First, the test number of orthogonal array in Taguchi's method is too much, which requires much experimentation and increases costs, and this is counter to its purpose of reducing costs. Second, the purpose of Taguchi's method is to reduce the effects of mutagenic factors rather than removing the mutagenic factors to improve quality. Third, despite the large amount of data, we are still unable to obtain any information about the interaction between controllable variable factors. Fourth, if there is no interaction between the controllable factors and interference factors, then a sound design does not exist. Therefore, it is necessary to find a simple method for optimizing sand casting performance parameters. A novel optimization method of performance parameters based on grey relational analysis was introduced here.

Grey relational analysis is a very active branch in grey system theory. Its basic idea is to determine the grey relational degree (GRD) between different sequences according to the geometrical shape of the sequence curves [15]. The larger the grey relational degree, the closer the project evaluated was to the ideal project. Accordingly, the order of the projects evaluated can be confirmed. Grey relational analysis does not require too many samples, nor do the samples have a typical distribution law, and the workload of calculation is relatively small. The grey relational analysis results are in good agreement with the qualitative analysis results. Grey relational analysis has been applied to many fields, such as decision-making [16], green supplier selection [17] and quality evaluation of red wine [18]. Wei [16] has investigated the dynamic hybrid multiple attribute decision-making problems based on grey relational analysis. In this study, grey relational analysis was used for optimization of sand casting performance parameters. During the process of grey relational analysis, assessment indicator weights should first be calculated.

Determination methods of indicator weights can be divided into subjective, objective and integrated weight methods. Subjective weight methods include the analytic hierarchy process [19] and Delphi method [20], which are mainly determined by experts' subjective cognition rather than objective data. Objective weight methods include principal components analysis [21], entropy weight method [22] and variation coefficient method [23], which are mainly confirmed by objective data. The integrated weight method [24–26] combines subjective and objective weight methods, reflecting experts' subjective cognition and objective data at the same time. The subjective and integrated weight methods need scoring by experts, which takes a lot of manpower and increases the workload. Therefore, the objective entropy weight method [22] was adopted in this study to calculate indicator weights.

The frequently used prediction techniques include back propagation (BP) neural network [27–29], grey system theory [30], principal component analysis [31], support vector machine [32] and Monte Carlo methods [33]. Among these prediction methods, BP neural network is a kind of nonlinear mathematical model that simulates nonlinear process of any degree. The biggest advantage of this model is that it can be trained and tested repeatedly, and finally it can approach any complicated nonlinear function. For example, Guo *et al.* [29] have proposed a hybrid wind speed-forecasting method based on BP neural network.

Once the optimization performance parameters were achieved, sand casting should refer to that batch to ensure safe casting and improve casting quality. But if some batches have data missing, they should be deleted and cannot be used for optimizing performance parameters, which will have a bad influence on optimizing performance parameters. What is more, the optimization performance parameters are less likely to be achieved due to shortage of enough valid data. Remeasurement of the missing data needs a lot of money, manpower and time, which would lead to a waste of resources. In addition, it is often difficult to find sand samples from the original casting batch due to the continuous production. Therefore, it was important and necessary here to predict the missing data based on the existing data. In the sand casting process, a portion of the recorded parameters data might be lost due to improper care, such as the tally book lost, or the storage data computer breakdown. Here, BP neural network [29] was introduced to predict missing data of sand casting. As the few papers published refer to the

basic principle of BP neural network, correlative theory of BP neural network was deduced to let readers have a better understanding of this model.

The main purpose of this paper was to optimize the sand casting performance parameters based on grey relational analysis; also, the missing data of sand casting can be predicted using BP neural network. To eliminate the influence of human factors, the objective entropy weight method was adopted in determining the weights of evaluation indicators. Before predicting missing data of sand casting, the relevant theory was deduced, and a new method was proposed for determining the number of hidden neurons in a network according to the mean square error of training samples.

# 2. Methods

This section mainly introduces basic principles of the composite model, including grey relational analysis, entropy weight method and BP neural network.

## 2.1. Grey relational analysis

Grey relational analysis is an important part of grey system theory [15]. The specific evaluation process of grey relational analysis is described as follows.

Let the original data matrix of the evaluation project be

$$Y = \begin{bmatrix} y_{11} & y_{12} & \cdots & y_{1n} \\ \cdots & \cdots & y_{ij} & \cdots \\ y_{m1} & y_{m2} & \cdots & y_{mn} \end{bmatrix},$$

where $m$ is the number of evaluation projects and $n$ the number of evaluation indicators; $y_{ij}$ ($1 \leq i \leq m$, $1 \leq j \leq n$) is the value of the $j$th evaluation indicator of the $i$th evaluation project.

Matrix $Y^* = [y_1 \ y_2 \ldots y_n]$ is the ideal project, where $y_j$ is the ideal value of the $j$th evaluation indicator. For positive indicators, the ideal value is the maximum; for negative indicators, it is the minimum; for moderate indicators, it is determined according to a particular case.

Because of different physical significance for each evaluation indicator, the evaluation indicators usually have different dimensions and order of magnitudes. Evaluation indicators should be dimensionless for comparison, reducing the interference of dimensions. The mean method was applied to non-dimensionalize the evaluation indicators, as shown in the following equation [16]:

$$z_{ij} = \frac{y_{ij}}{\sum_{i=1}^{m} y_{ij}/m} \quad j = 1, 2, \ldots, n, \tag{2.1}$$

where $z_{ij}$ is the dimensionless value of the $j$th evaluation indicator of the $i$th evaluation project.

After dimensionless, the original data matrix of evaluation indicators was transferred as follows:

$$Z = \begin{bmatrix} z_{11} & z_{12} & \cdots & z_{1n} \\ \cdots & \cdots & z_{ij} & \cdots \\ z_{m1} & z_{m2} & & z_{mn} \end{bmatrix},$$

and the ideal project $Z^* = [z_1^* \ z_2^* \ldots z_n^*]$, $z_n^*$ is the ideal dimensionless value of the $j$th evaluation indicator.

With dimensionless evaluation indicators, let the ideal project $Z^*$ be the reference sequence and the evaluation project $Z$ be the sequence to be compared, in which case the grey relational coefficient of the $j$th evaluation indicator of the $i$th evaluation project can be achieved according to the following equation [18]:

$$\xi_{ij} = \frac{\min\limits_{\substack{1 \leq i \leq m \\ 1 \leq j \leq n}} |z_j^* - z_{ij}| + 0.5 \times \max\limits_{\substack{1 \leq i \leq m \\ 1 \leq j \leq n}} |z_j^* - z_{ij}|}{|z_j^* - z_{ij}| + 0.5 \times \max\limits_{\substack{1 \leq i \leq m \\ 1 \leq j \leq n}} |z_j^* - z_{ij}|}, \tag{2.2}$$

where $\xi_{ij}$ is the grey relational coefficient of the $j$th evaluation indicator of the $i$th evaluation project.

Let the evaluation indicator weights of projects evaluated be $W = [w_1 \ w_2 \ldots w_n]$, in which case the GRD of projects evaluated can be obtained according to the following equation [17]:

$$r_i = \sum_{j=1}^{n} \xi_{ij} \times w_j \quad i = 1, 2, \ldots, m, \tag{2.3}$$

where $r_i$ is the GRD of the $i$th project evaluated.

The larger the $r_i$, the closer the $i$th project evaluated was to the ideal project $Z^*$. Accordingly, the order of the projects evaluated can be confirmed.

## 2.2. Entropy weight method

The procedures of the objective entropy weight method are presented below [22].

From the original data matrix $Y$, the data proportion was calculated based on the following equation:

$$p_{ij} = \frac{y_{ij}}{\sum_{i=1}^{m} y_{ij}} \quad j = 1, 2, \ldots, n, \tag{2.4}$$

where $p_{ij}$ is the data proportion of the $j$th evaluation indicator of the $i$th evaluation project.

The entropy of evaluation indicators was achieved according to the following equation:

$$e_j = -\frac{\sum_{i=1}^{m} p_{ij} \cdot \ln p_{ij}}{\ln m}, \tag{2.5}$$

where $e_j$ is entropy of the $j$th evaluation indicator.

The entropy weight of evaluation indicators was calculated based on the following equation:

$$w_j = \frac{1 - e_j}{n - \sum_{j=1}^{n} e_j}, \tag{2.6}$$

where $w_j$ is the entropy weight of the $j$th evaluation indicator.

According to the calculating process, the smaller the entropy $e_j$, the larger the variation in evaluation indicators, the more information it provided, and the larger weight it should be given.

## 2.3. BP neural network

In 1986, Parallel Distributed Procession, headed by Rumelhart and McClelland, published *Parallel distributed processing: exploration in the microstructures of cognition* [34]. They applied BP network to neural networks and started a new era of BP neural network research. Subsequently, BP neural networks have been widely studied and rapidly developed [35,36], and the relevant theory has been gradually improved. BP neural network consists of two parts, the forward propagation of information and back propagation (BP) of error.

During forward propagation, the information incoming from the input layer, after transition in the hidden layer, transfers to the output layer. When the actual and expected output values do not match, it will turn into error BP. The output error correction alters the weight and threshold value of each layer in a certain way, and back propagates to the hidden and input layers. The autonomic learning process of BP neural network involves the input, hidden and output layers constantly adjusting the weight and threshold values, which is also the process of forward information propagation and error BP. When the output error is less than the default, or the number of epochs reaches the preset value, the learning process is over.

BP neural network usually has three layers, the input, hidden and output layers (figure 1).

Let $x_j$ be the input value of an input layer neuron, $j = 1, 2, \ldots, m$ and $o_k$ the output value of the $k$th output layer neuron (figure 1).

In BP neural network, each layer contains several neurons. The input and output layers have one layer each and the hidden layer is set according to the structural features of the network as one or more layers; there might also not be a hidden layer.

### 2.3.1. Forward propagation of information

The input $net_i$ and output $o_i$ of the $i$th neuron of hidden layer are shown in the following equations:

$$\text{net}_i = \sum_{j=1}^{m} w_{ij} x_j + \theta_i \tag{2.7}$$

and

$$o_i = \phi(\text{net}_i), \tag{2.8}$$

where $net_i$ is the input of the $i$th neuron of hidden layer; $w_{ij}$ is the weight from the $i$th hidden layer neuron to the $j$th input layer neuron; $\theta_i$ is the threshold value of the $i$th hidden layer neuron; $o_i$ is the output of the $i$th neuron of hidden layer; $\phi$ is the active function of the hidden layer.

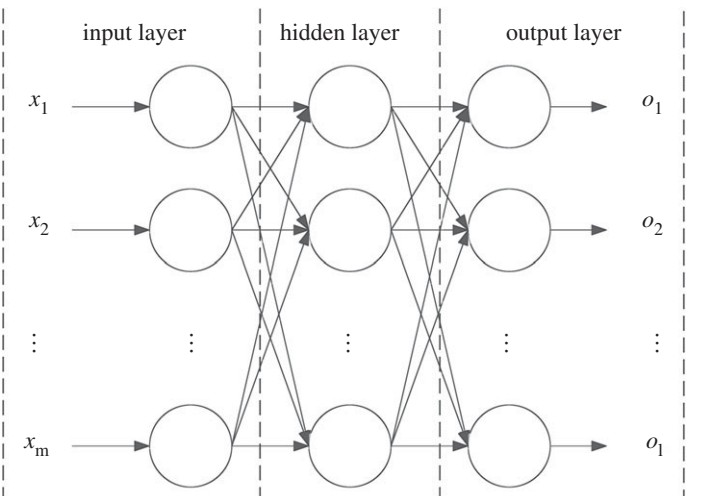

**Figure 1.** Structural chart of BP neural network.

The input $\mathrm{net}_k$ and output $o_k$ of the $k$th neuron of output layer are shown in the following equations:

$$\mathrm{net}_k = \sum_{i=1}^{q} w_{ki} o_i + a_k \tag{2.9}$$

and

$$o_k = \psi(\mathrm{net}_k), \tag{2.10}$$

where $\mathrm{net}_k$ is the input of the $k$th neuron of hidden layer; $w_{ki}$ is the weight from the $k$th output layer neuron to the $i$th hidden layer neuron, $i = 1, 2, \dots, q$; $a_k$ is the threshold value of the $k$th output layer neuron, $k = 1, 2, \dots, l$; $o_k$ is the output of the $k$th neuron of hidden layer; $\psi$ is the active function of the output layer.

### 2.3.2. Back propagation of error

During error BP, the output layer calculates the neuron output error of each layer, then adjusts the weight and threshold values of each layer according to the gradient descent method, finally letting the revised output values meet requirements.

The quadric-form error of each sample is shown in the following equation:

$$E_s = \frac{1}{2} \sum_{k=1}^{l} (T_k - o_k)^2, \tag{2.11}$$

where $s$ indicates the sample; $T_k$ is the expected output value of the sample at the $k$th neuron; $E_s$ is the quadric-form error of sample $s$.

The total error of the system is shown in the following equation:

$$E = \frac{1}{2} \sum_{s=1}^{p} \sum_{k=1}^{l} (T_k^s - o_k^s)^2, \tag{2.12}$$

where $p$ is the number of the samples; $E$ is the total error of the system.

The weight adjustment function of the output layer is shown in the following equation:

$$\Delta w_{ki} = -\eta \frac{\partial E}{\partial w_{ki}} = -\eta \frac{\partial E}{\partial o_k} \frac{\partial o_k}{\partial \mathrm{net}_k} \frac{\partial \mathrm{net}_k}{\partial w_{ki}}, \tag{2.13}$$

where $\eta$ is the learning rate; $\Delta w_{ki}$ is the weight adjustment function of the output layer.

The threshold value adjustment function of the output layer is shown in the following equation:

$$\Delta \alpha_k = -\eta \frac{\partial E}{\partial a_k} = -\eta \frac{\partial E}{\partial o_k} \frac{\partial o_k}{\partial \mathrm{net}_k} \frac{\partial \mathrm{net}_k}{\partial a_k}, \tag{2.14}$$

where $\Delta \alpha_k$ is the threshold value adjustment function of the output layer.

The weight adjustment function of the hidden layer is shown in the following equation:

$$\Delta w_{ij} = -\eta \frac{\partial E}{\partial w_{ij}} = -\eta \frac{\partial E}{\partial o_i} \frac{\partial o_i}{\partial \mathrm{net}_i} \frac{\partial \mathrm{net}_i}{\partial w_{ij}}, \tag{2.15}$$

where $\Delta w_{ij}$ is the weight adjustment function of the hidden layer.

The threshold value adjustment function of the hidden layer is shown in the following equation:

$$\Delta \theta_i = -\eta \frac{\partial E}{\partial \theta_i} = -\eta \frac{\partial E}{\partial o_i} \frac{\partial o_i}{\partial \mathrm{net}_i} \frac{\partial \mathrm{net}_i}{\partial \theta_i}, \tag{2.16}$$

where $\Delta \theta_i$ is the threshold value adjustment function of the hidden layer.

In equation (2.12), the total error derivation of $o_k$ is shown in the following equation:

$$\frac{\partial E}{\partial o_k} = -\sum_{s=1}^{p} \sum_{k=1}^{l} (T_k^s - o_k^s), \tag{2.17}$$

where $\partial E / \partial o_k$ is the total error derivation of $o_k$ in equation (2.12).

In equation (2.9), derivation of $w_{ki}$ and $a_k$ is respectively shown in the following equations:

$$\frac{\partial \mathrm{net}_k}{\partial w_{ki}} = o_i \tag{2.18}$$

and

$$\frac{\partial \mathrm{net}_k}{\partial a_k} = 1, \tag{2.19}$$

where $\partial \mathrm{net}_k / \partial w_{ki}$ is the derivative of equation (2.9) on $w_{ki}$; $\partial \mathrm{net}_k / \partial a_k$ is the derivative of equation (2.9) on $a_k$.

In equation (2.7), derivation of $w_{ij}$ and $\theta_i$ is respectively shown in equations (2.20) and (2.21).

$$\frac{\partial \mathrm{net}_i}{\partial w_{ij}} = x_j \tag{2.20}$$

and

$$\frac{\partial \mathrm{net}_i}{\partial \theta_i} = 1, \tag{2.21}$$

where $\partial \mathrm{net}_i / \partial w_{ij}$ is the derivative of equation (2.7) on $w_{ij}$; $\partial \mathrm{net}_i / \partial \theta_i$ is the derivative of equation (2.7) on $\theta_i$.

In equation (2.8), derivative of $\mathrm{net}_i$ is shown in the following equation:

$$\frac{\partial o_i}{\partial \mathrm{net}_i} = \phi'(\mathrm{net}_i), \tag{2.22}$$

where $\partial o_i / \partial \mathrm{net}_i$ is the derivative of equation (2.8) on $\mathrm{net}_i$.

In equation (2.10), derivative of $\mathrm{net}_k$ is shown in the following equation:

$$\frac{\partial o_k}{\partial \mathrm{net}_k} = \psi'(\mathrm{net}_k), \tag{2.23}$$

where $\partial o_k / \partial \mathrm{net}_k$ is the derivative of equation (2.10) on $\mathrm{net}_k$.

From equations (2.15) and (2.16), to achieve the weight and threshold value adjustment function, the function $\partial E / \partial o_i$ should be first obtained. From the above analysis, there was no direct contact between the system total error $E$ and neuron output $o_i$ of the hidden layer. Therefore, it was needed to first turn $\partial E / \partial o_i$ into equation (2.24).

$$\frac{\partial E}{\partial o_i} = \frac{\partial E}{\partial o_k} \frac{\partial o_k}{\partial \mathrm{net}_k} \frac{\partial \mathrm{net}_k}{\partial o_i}, \tag{2.24}$$

where $\partial E / \partial o_i$ is the derivative of system total error $E$ on neuron output $o_i$.

Equations (2.17) and (2.23) were input into equation (2.24), and the equation for $\partial E/\partial o_i$ was obtained, as shown in the following equation:

$$\frac{\partial E}{\partial o_i} = -\sum_{s=1}^{p}\sum_{k=1}^{l}(T_k^s - o_k^s)\cdot\psi'(\mathrm{net}_k)\cdot w_{ki}. \tag{2.25}$$

Equations (2.17), (2.23) and (2.18) were input into equation (2.13), and the weight adjustment function of the output layer was obtained, as shown in the following equation:

$$\Delta w_{ki} = \eta\sum_{s=1}^{p}\sum_{k=1}^{l}(T_k^s - o_k^s)\cdot\psi'(\mathrm{net}_k)\cdot o_i. \tag{2.26}$$

Equations (2.17), (2.23) and (2.19) were input into equation (2.14), and the threshold value adjustment function of the output layer was obtained, as shown in the following equation:

$$\Delta\alpha_k = \eta\sum_{s=1}^{p}\sum_{k=1}^{l}(T_k^s - o_k^s)\cdot\psi'(\mathrm{net}_k). \tag{2.27}$$

Equations (2.25), (2.22) and (2.20) were input into equation (2.15), and the weight adjustment function of the hidden layer was obtained, as shown in the following equation:

$$\Delta w_{ij} = \eta\sum_{s=1}^{p}\sum_{k=1}^{l}(T_k^s - o_k^s)\cdot\psi'(\mathrm{net}_k)\cdot w_{ki}\cdot\phi'(\mathrm{net}_i)\cdot x_j. \tag{2.28}$$

Equations (2.25), (2.22) and (2.21) were input into equation (2.16), and the threshold value adjustment function of the hidden layer was obtained, as shown in the following equation:

$$\Delta\theta_i = \eta\sum_{s=1}^{p}\sum_{k=1}^{l}(T_k^s - o_k^s)\cdot\psi'(\mathrm{net}_k)\cdot w_{ki}\cdot\phi'(\mathrm{net}_i). \tag{2.29}$$

### 2.3.3. Related functions of the training process

The performance function uses the mean squared error, as shown in the following equation (2.30):

$$\mathrm{mse} = \frac{1}{n}\sum_{i=1}^{n}(y_i - \overline{y_i})^2, \tag{2.30}$$

where mse is short for mean squared error; $n$ is the length of data predicted; $y_i$ is the actual value; $\overline{y_i}$ is the predicted value.

The active function of the hidden layer used a hyperbolic tangent sigmoid transfer function, as shown in the following equation (2.31):

$$\phi(x) = \frac{2}{1 + e^{-2x}} - 1, \tag{2.31}$$

where $x$ indicates the independent variable of active function of the hidden layer.

The active function of the output layer used purelin, a transfer function, as shown in the following equation:

$$\psi(u) = u, \tag{2.32}$$

where $u$ indicates the independent variable of active function of the output layer.

The training function used was trainlm, based on Levenberg–Marquardt BP.

The learning function used was learngd, based on the gradient descent method.

## 2.4. Procedure for model formulated

The main procedures for this composite model formulated based on grey relational analysis and BP neural network were as follows:

(1) Determining the weight of evaluation indicators. To avoid the influence of human factors, the weight of evaluation indicators was determined by the objective entropy weight method, which also saved a

**Table 1.** Sand casting performance parameters and their testing frequency.

| performance parameters | sampling spot | testing frequency |
|---|---|---|
| venting quality (VQ), wet compressive strength (WCS), moisture content (MC), compactability (Com) | discharge port of sand mill, or conveyer of foundry sand | once every half to two hours |
| | under the hopper of moulding machine | once every four to five hours |
| content of effective braize, content of effective bentonite, wet-heat tensile strength | under the hopper of moulding machine | once a day |
| content of clay, content of lump, grain composition | under the hopper of moulding machine | once a week |
| sand temperature, availability of bentonite, mobility, fracture and heat shock time | under the hopper of moulding machine | in case of need |

lot of manpower. The larger variation degree in the evaluation indicators indicated that the projects evaluated had good discrimination in this regard, and a larger weight should be given to these evaluation indicators.

(2) Selecting the optimization performance parameters of sand casting. The GRD can be achieved after the weight of evaluation indicators was determined. The larger the GRD, the closer the evaluated project was to the ideal project. Accordingly, the order of the projects evaluated was confirmed.

(3) Predicting the missing data of performance parameters. Dividing the performance parameters data into training samples and prediction samples. Using the training samples for training BP neural network, then predicting the missing data based on the trained BP neural network. This paper also provided a new method for determining the number of hidden neurons in a network.

# 3. Results

## 3.1. Selection of sand casting performance parameters

The foundry sand has many performance parameters, and each of them has an impact on the casting quality. However, the influence of each performance parameter on the casting quality is not the same, and the testing frequency is smaller for the performance parameter which is more important for improving the casting quality. The sand casting performance parameters and their testing frequency are shown in table 1.

As shown in table 1, the sand casting performance parameters, such as VQ, WCS, MC and Com, are the most important factors for improving the casting quality, and the testing frequency is also the smallest. Therefore, this study optimizes the performance parameters, such as VQ, WCS, MC and Com, for improving casting quality.

Taking 40 batches of floor sand in the casting line, the original data of performance parameters in this foundry sand sample were measured, including VQ, WCS, MC and Com (shown in electronic supplementary material, table S1) [37].

Among these performance parameters, VQ is the ability that gas to penetrate polymer material under a certain degree of pressure and time. The venting capacity of sand casting is increased not only by the riser and gas vent, but also by the VQ of the foundry sand. The VQ of foundry sand should not be too low to avoid the occurrence of boiling and pore defects in the casting process. But, the VQ of foundry sand should not be too high to prevent the molten metal from infiltrating into the porosity, which will cause rough surface or abreuvage of casting. Therefore, the VQ of the foundry sand needs to be within an appropriate range, and should not be too high or too low. For high-density moulding, the VQ of foundry sand should be high. For low- and medium-density moulding, the VQ of foundry sand should be low.

WCS refers to the ability of an object to resist external pressure under the saturated water condition. If the WCS of foundry sand is insufficient, the sand mould may be damaged or collapsed during the process of drawing and mould assembling; in the pouring process, the sand mould may not withstand the impact of molten metal, which will cause blisters or even molten metal discharging from the parting surface. However, the WCS of foundry sand is not the higher the better. The higher WCS of foundry sand needs more bentonite, which not only affects the MC and VQ of foundry sand,

**Table 2.** Grey relational coefficient of evaluation indicators.

| batch no. | VQ | WCS | MC | Com | batch no. | VQ | WCS | MC | Com |
|-----------|--------|--------|--------|--------|-----------|--------|--------|--------|--------|
| 1 | 0.5634 | 0.7557 | 0.8111 | 0.5721 | 21 | 0.8524 | 0.7364 | 0.7834 | 0.3607 |
| 2 | 0.8524 | 0.4953 | 0.6626 | 0.3607 | 22 | 0.3821 | 0.7557 | 0.945 | 0.9945 |
| 3 | 0.3358 | 0.7364 | 1 | 0.5721 | 23 | 0.5634 | 0.7364 | 0.945 | 0.499 |
| 4 | 0.8524 | 0.7557 | 0.9076 | 0.5721 | 24 | 0.4961 | 0.7364 | 0.9076 | 0.8091 |
| 5 | 0.5634 | 0.7557 | 0.7834 | 0.6702 | 25 | 0.7073 | 0.4953 | 0.7105 | 0.5638 |
| 6 | 0.4208 | 0.4953 | 0.945 | 0.7927 | 26 | 0.7073 | 0.7364 | 0.862 | 0.6702 |
| 7 | 0.5634 | 0.7364 | 0.8513 | 0.8091 | 27 | 0.5634 | 0.7557 | 0.7746 | 0.9945 |
| 8 | 0.7073 | 0.7557 | 0.7834 | 0.9945 | 28 | 0.8524 | 0.7364 | 0.945 | 0.6589 |
| 9 | 0.5634 | 0.4953 | 0.7105 | 0.5638 | 29 | 0.7073 | 0.7364 | 0.7179 | 0.5721 |
| 10 | 0.7073 | 0.7364 | 0.7746 | 0.5638 | 30 | 0.7073 | 0.7364 | 0.6562 | 0.4375 |
| 11 | 0.8524 | 0.4953 | 0.8957 | 0.7927 | 31 | 0.7073 | 0.7364 | 0.8957 | 0.5638 |
| 12 | 0.4961 | 0.7364 | 0.8208 | 0.6589 | 32 | 0.4961 | 0.4869 | 0.862 | 0.8091 |
| 13 | 0.8524 | 0.7557 | 0.9582 | 0.8091 | 33 | 0.7073 | 0.7364 | 0.7746 | 0.7927 |
| 14 | 0.7073 | 0.7557 | 0.8957 | 0.5638 | 34 | 0.7073 | 0.7364 | 0.8957 | 0.7927 |
| 15 | 0.5634 | 0.7557 | 0.7412 | 0.7927 | 35 | 0.8524 | 0.7557 | 0.862 | 0.9945 |
| 16 | 0.4961 | 0.7364 | 0.7834 | 0.7927 | 36 | 0.8524 | 0.7364 | 0.7834 | 0.9945 |
| 17 | 0.8524 | 0.7364 | 0.7492 | 0.9945 | 37 | 0.8524 | 0.7557 | 0.7105 | 0.6589 |
| 18 | 0.7073 | 0.7364 | 0.6151 | 0.8091 | 38 | 0.8524 | 0.4869 | 0.7105 | 0.5638 |
| 19 | 0.8524 | 0.7557 | 0.9582 | 0.7927 | 39 | 0.8524 | 0.7364 | 0.9582 | 0.8091 |
| 20 | 0.7073 | 0.7557 | 0.862 | 0.7927 | 40 | 0.7073 | 0.7364 | 0.9582 | 0.7927 |

but also increases the cost of casting. In addition, the higher WCS of foundry sand brings difficulties to the process of sand milling and shakeout.

MC is the percentage of water content to the total mass of the object. If the MC of foundry sand is low, the VQ of foundry sand will be high, and the casting is prone to sand burning. If the MC of foundry sand is high, a large amount of gas will be generated in the cavity due to evaporation of moisture during the pouring process. Once the gas in the cavity cannot be discharged smoothly within a limited time, an explosion accident may occur. Therefore, the MC of foundry sand should have a suitable range according to the filed practice.

Com is the proportion of the volume change of the object under a certain degree of pressure. On the one hand, the Com of foundry sand should not be too small; otherwise, the bentonite will be insufficiently wetted, leading the foundry sand to brittleness, low surface strength and difficulty in drawing. On the other hand, the Com of foundry sand should not be too large; otherwise, the castings are prone to boiling and pore defects.

In electronic supplementary material, table S1, it is assumed that all the performance parameters were within a reasonable range, meeting the normal production of sand casting. From the above analysis, it is known that all the performance parameters, namely VQ, WCS, MC and Com, belonged to the moderate indicator category, not too large or too small.

## 3.2. Optimization of performance parameters based on grey relational analysis

Sand casting performance parameters have a significant impact on casting quality, such that selection of the appropriate performance parameters was needed. The original data matrix was non-dimensionalized according to formula (2.1), and the ideal project can be achieved, where the optimal values of VQ, MC, Com and WCS were the maximum, minimum, average and maximum of the non-dimensionalized data, respectively.

The grey relational coefficient of evaluation indicators can be obtained based on formula (2.2), as shown in table 2.

**Table 3.** Weights of the evaluation indicators (using original data of evaluation indicators in electronic supplementary material, table S1).

| indicators | VQ | WCS | MC | Com |
|---|---|---|---|---|
| $e_j$ | 0.9496 | 0.9505 | 0.9501 | 0.9498 |
| $w_j$ | 0.252 | 0.2475 | 0.2495 | 0.251 |

**Table 4.** GRD of foundry sand.

| batch no. | GRD | batch no. | GRD | batch no. | GRD | batch no. | GRD |
|---|---|---|---|---|---|---|---|
| 1 | 0.675 | 11 | 0.7598 | 21 | 0.6831 | 31 | 0.7255 |
| 2 | 0.5932 | 12 | 0.6774 | 22 | 0.7687 | 32 | 0.6637 |
| 3 | 0.66 | 13 | 0.844 | 23 | 0.6853 | 33 | 0.7527 |
| 4 | 0.7719 | 14 | 0.7303 | 24 | 0.7368 | 34 | 0.7829 |
| 5 | 0.6927 | 15 | 0.7129 | 25 | 0.6196 | 35 | 0.8665 |
| 6 | 0.6634 | 16 | 0.7017 | 26 | 0.7438 | 36 | 0.8421 |
| 7 | 0.7397 | 17 | 0.8336 | 27 | 0.7719 | 37 | 0.7445 |
| 8 | 0.8104 | 18 | 0.7171 | 28 | 0.7982 | 38 | 0.6541 |
| 9 | 0.5833 | 19 | 0.8399 | 29 | 0.6832 | 39 | 0.8392 |
| 10 | 0.6953 | 20 | 0.7793 | 30 | 0.634 | 40 | 0.7985 |

In order to get the GRD of evaluation indicators, their weights should be known first. The weights of the evaluation indicators, including VQ, WCS, MC and Com, were achieved by entropy weight method based on formulae (2.4)–(2.6), as shown in table 3.

The GRD of foundry sand was achieved based on formulae (2.3) and the weights of the evaluation indicators (table 4).

As shown in table 4, the GRD of sample batch 35 was the largest, which was the closest to the ideal project. Thus, it was advised that the performance parameters of sand casting refer to batch 35 to ensure safe casting and improve casting quality.

## 3.3. Prediction of missing data using BP neural network

In this section, the data were divided into training samples and prediction samples. The first 39 batches of performance parameters from sand castings were training samples and the 40th batch the prediction sample. WCS, MC and Com were the input data and VQ the output data.

Any nonlinear function can be achieved by a three-layer BP neural network according to Kolmogorov theory. Here, the input layer contained three neurons and the output layer contained one neuron. Some scholars have studied the number of neurons in the hidden layer [38–41], but they have not reached consistent conclusions. The number of suitable neurons in the hidden layer was first examined, with the relationship between the mean square error of training samples and the number of hidden neurons, as shown in figure 2.

The mean square error of the training samples showed a downward trend as a whole when number of hidden neurons belonged to (1,9), but when it was greater than 9, the mean square error of the training samples basically remained unchanged (figure 2). Therefore, to simplify the calculation process and improve calculation efficiency, the number of hidden neurons was 9 in this network.

With the number of hidden neurons at 9, retraining the samples and the relationship between mean square error of training samples and epochs is shown in figure 3.

As the number of epochs increases, the mean square error decreases gradually, and it was the smallest at epoch 140 (figure 3). As the training sample data were not very good, the mean square error did not reach the goal.

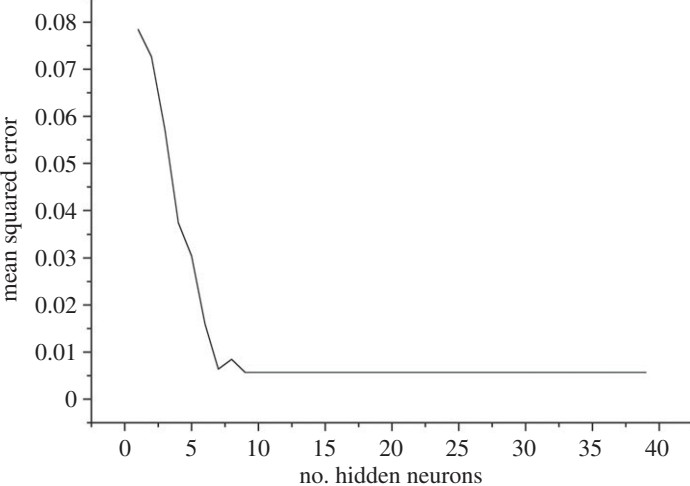

**Figure 2.** Relationship between mean square error and number of hidden neurons.

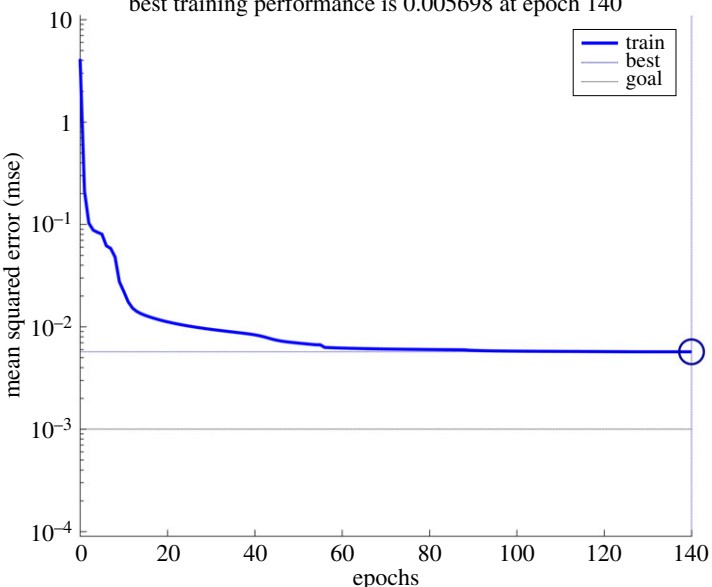

**Figure 3.** Relationship between mean square error and epoch.

The relationship between the output and target of samples during training is shown in figure 4.

As shown in figure 4, the little circle refers to the actual fitting data, the blue line represents the function relationship between target and output of training samples, and the imaginary line represents the output was equal to the target. The function relationship between target and output of training samples is shown in the following formula (3.1).

$$Output \approx 0.96 \times Target + 0.0033, \tag{3.1}$$

where *Output* indicates the output of samples during training; *Target* indicates the target of samples during training.

The correlation coefficient of fitting function was $R = 0.98037$. The output and target of the training samples were basically equal (figure 4). But due to poor correspondence between the training samples, the output and target of individual samples still had great error, which was consistent with the analysis results shown in figure 3.

Using the trained BP neural network to predict missing data, the sand sample WCS, MC and Com of the 40th batch were the input data and VQ the output data. The predictive result of VQ was 179.9028 and the relative error 0.054%. The prediction precision was thus good, which met engineering needs and can be used to predict missing data of sand casting.

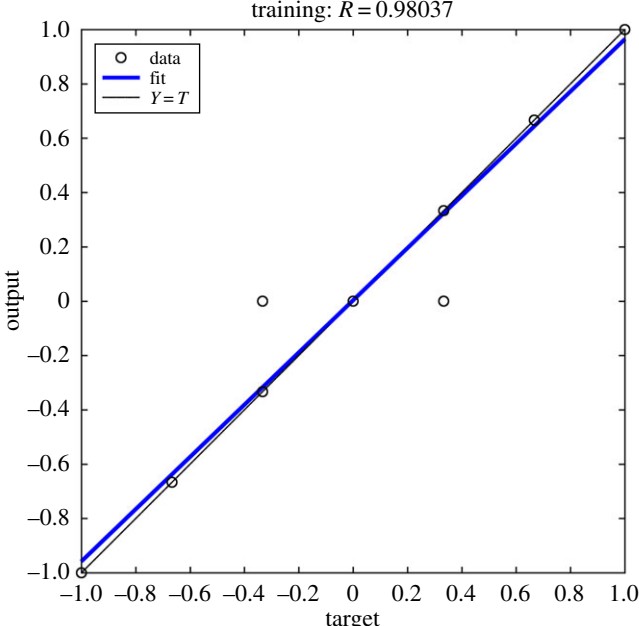

training: $R = 0.98037$

**Figure 4.** Regression analysis of training samples.

As each initialization possessed a random weight and threshold value, the prediction results were different each time. BP neural network with good training results can be saved and used for later predictions.

# 4. Discussion

## 4.1. Compared the predictive result with grey system theory

To demonstrate the validity and feasibility of BP neural network adopted in the process of missing data prediction, this section compares the result of missing data prediction with GM(1,1) model. GM(1,1) model is one basic and important part in the grey system theory, which refers to first-order Grey Model in one variable [20]. The brief procedure of GM(1,1) model was as follows [20].

Let the original data sequence be $X^0 = (x^{(0)}(1), x^{(0)}(2), \ldots, x^{(0)}(n))$ and the accumulation data sequence $X^1 = (x^{(1)}(1), x^{(1)}(2), \ldots, x^{(1)}(n))$, where $x^{(1)}(i)$ can be calculated based on the following formula:

$$x^{(1)}(k) = \sum_{i=1}^{k} x^{(0)}(i), k = 1, 2, \ldots, n, \tag{4.1}$$

where $x^{(0)}(i)$ indicates the original data; $x^{(1)}(k)$ indicates the accumulation data.

Suppose the matrix $X^{(1)}$ accords with the exponential change law, and the whitenization equation of GM(1,1) model is as follows:

$$\frac{\mathrm{d}x^{(1)}}{\mathrm{d}t} + ax^{(1)} = b, \tag{4.2}$$

where $t$ indicates the time; $a$ indicates the development coefficient; $b$ indicates the grey action.

Let $\bar{x}^{(1)}(1) = x^{(1)}(1)$ be the initial conditions, solve the equation (4.2) and the predictive formula of $X^{(1)}$ can be achieved, as shown in (4.3).

$$\bar{x}^{(1)}(k+1) = \left[x^{(0)}(1) - \frac{b}{a}\right]e^{-ak} + \frac{b}{a}, \quad k = 0, 1, 2, \ldots, \tag{4.3}$$

where $\bar{x}^{(1)}(k+1)$ indicates the predictive value of accumulation data.

The predictive formula $X^{(0)}$ of the original data sequence is shown in the following formula (4.4), calculated as $x^{(1)}(k+1) - x^{(1)}(k)$:

$$\bar{x}^{(0)}(k+1) = (1 - e^{a})\left[x^{(0)}(1) - \frac{b}{a}\right]e^{-ak}, \quad k = 1, 2, 3, \ldots, \tag{4.4}$$

where $\bar{x}^{(0)}(k+1)$ indicates the predictive value of original data.

The development coefficient $a$ and grey action $b$ are based on the least squares estimate of GM(1,1) model, as shown in the following formula:

$$\bar{a} = (B^T B)^{-1} B^T Y = (a, b)^T,$$ 
(4.5)

where the matrix $B$ and $Y$ are as follows.

The background value $Z^{(1)}$ is the mean sequence of $X^1$, as calculated by the following formula:

$$Z^{(1)}(k+1) = \frac{1}{2}[X^{(1)}(k+1) + X^{(1)}(k)], \quad k = 1, 2, \ldots, n-1,$$ 
(4.6)

where $Z^{(1)}(k+1)$ indicates the background value.

Suppose VQ performance parameter of the first 39 batches were the original data sequence, then the predictive formula of VQ can be achieved based on the above procedures, as shown in the following equation:

$$\bar{x}^{(0)}(k+1) = 173.2346e^{0.001k}, \quad k = 1, 2, 3, \ldots.$$ 
(4.7)

Therefore, the VQ predictive value of the 40th batch was 180.1235 according to formula (4.7), and the relative error was 0.069%.

It can be seen that the predictive value of BP neural network was more precise than grey system theory. It is mainly because that BP neural network takes advantage of more information, and BP neural network is a kind of nonlinear mathematical model that can simulate nonlinear process of any degree. The biggest advantage of this model is that it can be trained and tested repeatedly, and finally it can approach any complicated nonlinear function.

## 4.2. Discussion of results

The present results confirmed that the composite model proposed in this paper was successfully applied to sand casting. This was the first report in this field of the optimization of performance parameters based on grey relational analysis and predictions of missing data based on BP neural network. The advantages of the composite model proposed in this paper were as follows. First, optimization performance parameters can contribute to reducing foundry defects and improving casting quality as well as ensuring safety in sand casting. Second, prediction of missing data can avoid repetitive waste of resources and the difficulties of retesting. Third, this paper provided a new method for determining the number of hidden neurons in a network according to the mean square error of training samples.

Motivated by previous studies on grey relational analysis [15–17], this method was introduced into the field of sand casting for performance parameters optimizing for the first time in this paper. The larger the GRD, the closer the project evaluated was to the ideal project. Accordingly, the order of the projects evaluated was confirmed. The GRD of sample batch 35 was the largest, so sample batch 35 was the optimization performance parameter of sand casting. The results showed that grey relational analysis can be easily used for performance parameters optimization of sand casting. Different from Taguchi's method [14], which requires a lot of experimentation, grey relational analysis is less demanding on the quantity and regularity of samples and can be easily calculated without any discrepancy between calculated results and quantitative analysis results. In the process of determining assessment indicators weights, to eliminate the influence of human factors, such as subjective [19,20] and integrated [24–26] weight methods, the objective entropy weight method [22] was adopted, which also significantly decreased manpower.

In determining the number of hidden neurons, there were not consistent results [38–41]. This paper provided a new method for determining the number of hidden neurons according to the mean square error of training samples. The results showed that mean square error of training samples was the smallest when the number of hidden neurons was 9 and the epoch 140 when training BP neural network, and the relative error of the prediction samples was 0.054%. BP neural network is a widely used prediction technique, but there are a few studies that refer to the basic principle [27–29], which is not conducive to the improvement of the algorithm. Therefore, relevant theory of BP neural network was deduced before predicting missing data, such that there will be a general understanding of the prediction principle. To demonstrate the validity and feasibility of BP neural network adopted in the process of missing data prediction, grey system theory [20] was applied to compare the result of missing data prediction. The relative error of predictive value was 0.069% based on grey system theory. The results showed that the predictive value of BP neural network showed more precision than grey system theory. The composite model proposed in this paper can be used for related research in this field.

# 5. Conclusion

A composite optimization and prediction model of sand casting performance parameters was proposed based on grey relational analysis and BP neural network. The main conclusions were as follows.

First, to avoid the influence of human factors, the weights of evaluation indicators were achieved based on the objective entropy weight method and the results were 0.252, 0.2475, 0.2495 and 0.251 for the VQ, WCS, MC and Com, respectively. The weight of performance parameter VQ was the biggest, indicating that the sand batch had good discrimination in this regard.

Second, the GRDs of foundry sand were obtained according to grey relational analysis after the weight of performance parameters was determined, and the results showed that sample batch 35 possessed the largest GRD. Thus, it was advised that performance parameters of sand castings refer to batch 35 to ensure safety casting and improve casting quality.

Third, the performance parameters data were divided into training samples and prediction samples. The mean square error of training samples was the smallest when the number of hidden neurons was 9 and the epoch 140 when training BP neural network, and the relative error of the prediction samples was 0.054% based on this trained BP neural network. The relevant theory was deduced before predicting missing data, such that there will be a general understanding regarding the prediction principle of BP neural network.

Fourth, to demonstrate the validity and feasibility of BP neural network adopted in the process of missing data prediction, grey system theory was applied to compare the result of missing data prediction. The relative error of predictive value was 0.069% based on grey system theory. The results showed that the predictive value of BP neural network showed more precision than grey system theory.

Data accessibility. The original data of sand casting performance parameters and Matlab code supporting this paper have been uploaded as electronic supplementary material.

Authors' contributions. K.X. and X.Y. guided the writing process, Q.X. and L.L. conducted the numerical experiments and data analysis and Q.X. wrote the paper. All the authors gave their final approval for publication.

Competing interests. We declare we have no competing interests.

Funding. This study was supported by the National Key Research and Development Program of China (grant no. 2017YFC0805100) and Basic Scientific Research Program of China Academy of Safety Science and Technology (grant no. 2017JBKY12).

Acknowledgements. We thank the editor and anonymous reviewers for their valuable comments and suggestions.

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
