## [Reviewer comments · Royal Society Open Science]

Review History

RSOS-180065.R0 (Original submission)

Review form: Reviewer 1

Is the manuscript scientifically sound in its present form?

No

Are the interpretations and conclusions justified by the results?

No

Is the language acceptable?

Yes

Is it clear how to access all supporting data?

No

Do you have any ethical concerns with this paper?

No

Have you any concerns about statistical analyses in this paper?

No

Recommendation?

Reject

Comments to the Author(s)

Dear Author(s)

Ref: Submission Optimization of sand-casting process parameters and missing data prediction

Thank you for submitting your work to the International Journal of Royal Society Open Science..

We regretfully conclude that sentences in this paper seem to have been borrowed from other published articles.

We do thank you for your interest in the International Journal of Royal Society Open Science. There are the some points, which are to be taken care of.

1. On Page No. 2 line No. 25-28: is a contradictory statement, how can data be missed and would lead to waste of resources.
2. Some of the statement in "Introduction" are not relevant e.g. line No. 31-40 etc. on Page 2.
3. No source is given for all the formulas.
4. Most of the Abbreviations are not defined.
5. After giving a small description of methods, jumped to results. No problem formulation is given
6. There is no range given for the process parameters for optimization.
7. On Page 10 line No 18, GP, GCS "positive indicator" and for MC "negative indicator" on which base, how it is being decided. Line No. 21-23 self contradictory.
8. Figure No. 4 is not explained.
9. Model is not formulated.
10. Conclusion is not self explanatory.
11. Some of the references are not referred at all in the text and some are not even relevant to topic itself.

I have looked through your paper, and I have to decline it.

Respectfully,

Best Regards,

Review form: Reviewer 2

Is the manuscript scientifically sound in its present form?

No

Are the interpretations and conclusions justified by the results?

No

Is the language acceptable?

No

Is it clear how to access all supporting data?

Not Applicable

Do you have any ethical concerns with this paper?

No

Have you any concerns about statistical analyses in this paper?

No

Recommendation?

Major revision is needed (please make suggestions in comments)

Comments to the Author(s)

To optimize the process parameters in sand casting, a method based on gray relational analysis is presented and BP neural network is used to predict missing data in this paper.

I have the following major concerns regarding this paper.

1. There are a lot of issues on the language expression, some sentence patterns are not used correctly.
2. The relationship between parameters optimization and data missing prediction can't be found. It seems the two key points proposed in this paper are fragmented.
3. Both gray relational analysis and BP neural network are quite conventional methods. This paper devotes much space (about 7 pages) describing the basic principle of common methods, and there is little improvement.
4. From my point of view, BP neural network is a simple but not an effective method for missing data prediction in this paper. In fact, people are not interested in the process of network training and its parameter selection. It's meaningless for common methods like BP NN. What is the advantage of the method proposed in this paper? There should be more comparison between different missing data predictions.
5. In chapter 4. Discussion, there should be deeper analysis but not similar content like overview.

Decision letter (RSOS-180065.R0)

01-Feb-2018

Dear Dr Xu:

Manuscript ID RSOS-180065 entitled "Optimization of sand-casting process parameters and missing data prediction" which you submitted to Royal Society Open Science, has been reviewed. The comments from reviewers are included at the bottom of this letter.

In view of the criticisms of the reviewers, the manuscript has been rejected in its current form. However, a new manuscript may be submitted which takes into consideration these comments.

Please note that resubmitting your manuscript does not guarantee eventual acceptance, and that your resubmission will be subject to peer review before a decision is made.

Once you have revised your manuscript, go to <https://mc.manuscriptcentral.com/rsos> and login

to your Author Center. Click on "Manuscripts with Decisions," and then click on "Create a Resubmission" located next to the manuscript number. Then, follow the steps for resubmitting your manuscript.

Your resubmitted manuscript should be submitted by 01-Aug-2018. If you are unable to submit by this date please contact the Editorial Office.

Please note that Royal Society Open Science will introduce article processing charges for all new submissions received from 1 January 2018. Charges will also apply to papers transferred to Royal Society Open Science from other Royal Society Publishing journals, as well as papers submitted as part of our collaboration with the Royal Society of Chemistry (<http://rsos.royalsocietypublishing.org/chemistry>). If your manuscript is submitted and accepted for publication after 1 Jan 2018, you will be asked to pay the article processing charge, unless you request a waiver and this is approved by Royal Society Publishing. You can find out more about the charges at <http://rsos.royalsocietypublishing.org/page/charges>. Should you have any queries, please contact openscience@royalsociety.org.

on behalf of Professor Jun Fu (Associate Editor) and R. Kerry Rowe (Subject Editor)
openscience@royalsociety.org

Associate Editor Comments to Author (Professor Jun Fu):

The reviewers pointed out the lack of scientific novelty and effectiveness of significant techniques in this manuscript. After my carefully reading, I concur with reviewers and thus my recommendation is Reject and allow resubmission.

Reviewers' Comments to Author:
Reviewer: 1

Comments to the Author(s)
Dear Author(s)

Ref: Submission Optimization of sand-casting process parameters and missing data prediction

Thank you for submitting your work to the International Journal of Royal Society Open Science.

We regretfully conclude that sentences in this paper seem to have been borrowed from other published articles.

We do thank you for your interest in the International Journal of Royal Society Open Science. There are the some points, which are to be taken care of.

1. On Page No. 2 line No. 25-28: is a contradictory statement, how can data be missed and would lead to waste of resources.

2. Some of the statement in "Introduction" are not relevant e.g. line No. 31-40 etc. on Page 2.
3. No source is given for all the formulas.
4. Most of the Abbreviations are not defined.
5. After giving a small description of methods, jumped to results. No problem formulation is given
6. There is no range given for the process parameters for optimization.
7. On Page 10 line No 18, GP, GCS "positive indicator" and for MC "negative indicator" on which base, how it is being decided. Line No. 21-23 self contradictory.
8. Figure No. 4 is not explained.
9. Model is not formulated.
10. Conclusion is not self explanatory.
11. Some of the references are not referred at all in the text and some are not even relevant to topic itself.

I have looked through your paper, and I have to decline it.

Respectfully,
Best Regards,

Reviewer: 2

Comments to the Author(s)

To optimize the process parameters in sand casting, a method based on gray relational analysis is presented and BP neural network is used to predict missing data in this paper.

I have the following major concerns regarding this paper.

1. There are a lot of issues on the language expression, some sentence patterns are not used correctly.
2. The relationship between parameters optimization and data missing prediction can't be found. It seems the two key points proposed in this paper are fragmented.
3. Both gray relational analysis and BP neural network are quite conventional methods. This paper devotes much space (about 7 pages) describing the basic principle of common methods, and there is little improvement.
4. From my point of view, BP neural network is a simple but not an effective method for missing data prediction in this paper. In fact, people are not interested in the process of network training and its parameter selection. It's meaningless for common methods like BP NN. What is the advantage of the method proposed in this paper? There should be more comparison between different missing data predictions.
5. In chapter 4. Discussion, there should be deeper analysis but not similar content like overview.

Author's Response to Decision Letter for (RSOS-180065.R0)

See Appendix A.

RSOS-180349.R0

Review form: Reviewer 3

Is the manuscript scientifically sound in its present form?

Yes

Are the interpretations and conclusions justified by the results?

Yes

Is the language acceptable?

No

Is it clear how to access all supporting data?

Yes

Do you have any ethical concerns with this paper?

No

Have you any concerns about statistical analyses in this paper?

No

Recommendation?

Major revision is needed (please make suggestions in comments)

Comments to the Author(s)

The English need revisions, and the process of optimization needs clarification, there is no information about the application of entropy to the available data

Decision letter (RSOS-180349.R0)

08-Oct-2018

Dear Dr Xu:

Manuscript ID RSOS-180349 entitled "Optimization of sand casting process parameters and missing data prediction" which you submitted to Royal Society Open Science, has been reviewed. The comments from reviewer(s) are included at the bottom of this letter.

In view of the criticisms of the reviewer(s), I must decline the manuscript for publication in Royal Society Open Science at this time. However, a new manuscript may be submitted which takes into consideration these comments.

Please note that resubmitting your manuscript does not guarantee eventual acceptance, and that your resubmission will be subject to re-review by the reviewer(s) before a decision is rendered.

You will be unable to make your revisions on the originally submitted version of your manuscript. Instead, revise your manuscript using a word processing program and save it on your computer.

You may also click the below link to start the resubmission process (or continue the process if you

have already started your resubmission) for your manuscript. If you use the below link you will not be required to login to ScholarOne Manuscripts.

*** PLEASE NOTE: This is a two-step process. After clicking on the link, you will be directed to a webpage to confirm. ***

https://mc.manuscriptcentral.com/rsos?URL_MASK=140880b8211c4ec8bd0d7df9c31953ea

Because we are trying to facilitate timely publication of manuscripts submitted to Royal Society Open Science, your resubmitted manuscript should be submitted by 07-Apr-2019. If you are unable to submit by this date please contact the Editorial Office for options.

Please note that Royal Society Open Science will introduce article processing charges for all new submissions received from 1 January 2018. Charges will also apply to papers transferred to Royal Society Open Science from other Royal Society Publishing journals, as well as papers submitted as part of our collaboration with the Royal Society of Chemistry (<http://rsos.royalsocietypublishing.org/chemistry>). If your manuscript is submitted and accepted for publication after 1 Jan 2018, you will be asked to pay the article processing charge, unless you request a waiver and this is approved by Royal Society Publishing. You can find out more about the charges at <http://rsos.royalsocietypublishing.org/page/charges>. Should you have any queries, please contact openscience@royalsociety.org.

I look forward to a resubmission.

on behalf of Professor Jun Fu (Associate Editor) and Professor R. Kerry Rowe (Subject Editor)
openscience@royalsociety.org

Associate Editor Comments to Author (Professor Jun Fu):

A reviewer pointed out that this manuscript lacked the information about the application of entropy to the available data and the clarification for the process of optimization. Additionally, sentences in this paper seem to have been borrowed from other published articles according to the result of iThenticate for plagiarism checking. Thus my recommendation is Reject & allow resubmission.

Reviewer comments to Author:
Reviewer: 3

Comments to the Author(s)
the English need revisions, and the process of optimization needs clarification, there is no information about the application of entropy to the available data

Author's Response to Decision Letter for (RSOS-180349.R0)

See Appendix B.

RSOS-181860.R0

Review form: Reviewer 4

Is the manuscript scientifically sound in its present form?

Yes

Are the interpretations and conclusions justified by the results?

Yes

Is the language acceptable?

No

Is it clear how to access all supporting data?

Yes

Do you have any ethical concerns with this paper?

No

Have you any concerns about statistical analyses in this paper?

I do not feel qualified to assess the statistics

Recommendation?

Accept with minor revision (please list in comments)

Comments to the Author(s)

Clarity of equations and figures (Like Figure 1, 3, 4) need to be improved.

Explanation is required for each equation to improve the readability of the paper.

Original data set need to be added as Appendix

Review form: Reviewer 5 (Manickam Ramachandran)

Is the manuscript scientifically sound in its present form?

No

Are the interpretations and conclusions justified by the results?

No

Is the language acceptable?

Yes

Is it clear how to access all supporting data?

Not Applicable

Do you have any ethical concerns with this paper?

No

Have you any concerns about statistical analyses in this paper?

Yes

Recommendation?

Major revision is needed (please make suggestions in comments)

Comments to the Author(s)

Review Comments

1. Give more importance for the experiment and not on the basis of the optimization tool
2. More justification needed on why this optimization tool is used and why not others.
3. Why authors have chosen the factors of performance parameters and what all other possible performance parameters in sand casting explain in details.

Decision letter (RSOS-181860.R0)

04-Jun-2019

Dear Dr Xu,

The Subject Editor assigned to your paper ("Optimization of sand casting performance parameters and missing data prediction") has now received comments from reviewers. We would like you to revise your paper in accordance with the referee and Associate Editor suggestions which can be found below (not including confidential reports to the Editor). Please note this decision does not guarantee eventual acceptance.

Please submit a copy of your revised paper before 27-Jun-2019. Please note that the revision deadline will expire at 00.00am on this date. If we do not hear from you within this time then it will be assumed that the paper has been withdrawn. In exceptional circumstances, extensions may be possible if agreed with the Editorial Office in advance. We do not allow multiple rounds of revision so we urge you to make every effort to fully address all of the comments at this stage. If deemed necessary by the Editors, your manuscript will be sent back to one or more of the original reviewers for assessment. If the original reviewers are not available we may invite new reviewers.

When submitting your revised manuscript, you must respond to the comments made by the referees and upload a file "Response to Referees" in "Section 6 - File Upload". Please use this to document how you have responded to each of the comments, and the adjustments you have made. In order to expedite the processing of the revised manuscript, please be as specific as possible in your response.

- Ethics statement

- Data accessibility

<http://datadryad.org/submit?journalID=RSOS&manu=RSOS-181860>

- Competing interests

- Authors' contributions

- Acknowledgements

- Funding statement

on behalf of R. Kerry Rowe (Subject Editor)
openscience@royalsociety.org

Reviewer comments to Author:
Reviewer: 4

Comments to the Author(s)
Clarity of equations and figures (Like Figure 1, 3, 4) need to be improved.
Explanation is required for each equation to improve the readability of the paper.
Original data set need to be added as Appendix

Reviewer: 5

Comments to the Author(s)
Review Comments
1. Give more importance for the experiment and not on the basis of the optimization tool
2. More justification needed on why this optimization tool is used and why not others.
3. Why authors have chosen the factors of performance parameters and what all other possible performance parameters in sand casting explain in details.

Author's Response to Decision Letter for (RSOS-181860.R0)

See Appendix C.

RSOS-181860.R1 (Revision)

Review form: Reviewer 5 (Manickam Ramachandran)

Is the manuscript scientifically sound in its present form?

Yes

Are the interpretations and conclusions justified by the results?

Yes

Is the language acceptable?

Yes

Do you have any ethical concerns with this paper?

No

Recommendation?

Accept as is

Comments to the Author(s)

Nice work done

Decision letter (RSOS-181860.R1)

15-Jul-2019

Dear Dr Xu,

I am pleased to inform you that your manuscript entitled "Optimization of sand casting performance parameters and missing data prediction" is now accepted for publication in Royal Society Open Science.

Kind regards,

Andrew Dunn

on behalf of Professor Jun Fu (Associate Editor) and R. Kerry Rowe (Subject Editor)

Reviewer comments to Author:

Reviewer: 5

Comments to the Author(s)

Nice work done

Appendix A

RE: RSOS-180065

Dear Editor,

We thank you and the anonymous reviewers for giving us a constructive suggestions which will help us to improve the quality of the paper. Here we submit a new version of our manuscript, which has been modified according to the reviewers' suggestions.

The point to point responds to the reviewers' comments are listed as following, and we also setting up procedure of partial content.

To Dr Power

Response: Thank you for the comments in the E-mail. As you also state as follow.

Please do NOT deposit your manuscript figure or table files in the Dryad repository (<http://datadryad.org/>) -- this may cause unnecessary delays. Only datasets, code, or other digital research materials not already reported in your manuscript or included as electronic supplementary information should be included in the Dryad repository.

All relevant figures and tables are provided in the paper, therefore we upload the code as the electronic supplementary materials, and revise the data accessibility as follow

Data accessibility. The code that supporting this paper has been uploaded as electronic supplementary materials.

Reviewer: 1

Comment 1: On Page No. 2 line No. 25-28: is a contradictory statement, how can data be missed and would lead to waste of resources.

Response: Thank you for the comments on the paper. It's our fault that wish to express as much content as possible in one sentence, which caused a misunderstanding. The recorded parameters data might be lost due to improper care, such as the tally book lost, or the storage data computer breakdown. Remeasurement of the missing data needs a lot of money, manpower and time, which would lead to a waste of resources. Therefore, those sentences are revised as follows:

In the sand casting process, a portion of the recorded parameters data might be lost due to improper care, such as the tally book lost, or the storage data computer breakdown.

Remeasurement of the missing data needs a lot of money, manpower and time, which would lead to a waste of resources. In addition, it is often difficult to find sand samples from the original casting batch due to the continuous production. Therefore, it was important and necessary here to predict the missing data based on existing data.

Comment 2: Some of the statement in "Introduction" are not relevant e.g. line No. 31-40 etc. on Page 2.

Response: Thank you for your instructive suggestions. Partial content on page 2 line 31-40 does not have much relevance indeed, so we delete some content and revise those sentences by logic as follows: The frequently used prediction techniques include BP neural network [27-29], gray system theory [30], principal component analysis [31], support vector machine [32], and Monte Carlo methods [33]. Among these prediction methods, BP neural network is a kind of nonlinear mathematical model that simulates nonlinear process of any degree. The biggest advantage of this model is that it can be trained and tested repeatedly, and finally it can approach any complicated nonlinear function. For example, Guo [29] has proposed a hybrid wind speed-forecasting method based on BP neural network.

Comment 3: No source is given for all the formulas.

Response: Thank you very much to point out this question in our manuscript. We are sorry for this mistake. The formulas sources are added in the Gray relational analysis section as follows:

The mean method was applied to nondimensionalize the evaluation indicators, shown in formula 1 [16].
(1)

After dimensionless, the original data matrix of evaluation indicators was transferred as follows.

and the ideal project be .

With dimensionless evaluation indicators, let the ideal project be the reference sequence and the evaluation project be the sequence to be compared, in which case the gray relational coefficient of the jth evaluation indicator of the ith evaluation project can be achieved according to formula 2 [18].

(2)

The significant difference between gray relational coefficients was increased and might result in data distortion, which caused by the maximum absolute value being too large. To avoid this situation, the

resolution coefficient was introduced in formula 2, and usually set to .

Let the evaluation indicator weights of projects evaluated be , in which case the GRD of projects evaluated was obtained according to formula 3 [17].

(3)
The larger the GRD , the closer the i th project evaluated was to the ideal project . Accordingly, the order of the projects evaluated was confirmed.

The formulas sources are added in the Entropy weight method section as follows:

The procedure of the objective entropy weight method was presented below [22].

From the original data matrix Y , the data proportion was calculated based on formula 4.

(4)
The entropy of evaluation indicators was achieved according to formula 5.

(5)
The entropy weight of evaluation indicators was calculated based on formula 6.

(6)
From the calculating process, the smaller the entropy , the larger the variation in evaluation indicators, the more information it provided, and the larger weight it should be given.

The formulas in the BP neural network section are mainly deduced by ourselves, therefore there is few source in this part.

Comment 4: Most of the Abbreviations are not defined.

Response: Thank you for your valuable comments. The abbreviations that don't well known are defined in the paper as follows:

Among these process parameters, VQ is the ability that gas to penetrate high molecular material under a certain degree of pressure and time; WCS refers to the ability of object to resist external pressure under the saturated water condition; MC is the percentage of water content to the total mass of the object; Com is the proportion of the volume change of the object under a certain degree of pressure. GRD refers to the relational degree between the project evaluated and the ideal project.

Comment 5: After giving a small description of methods, jumped to results. No problem formulation is given

Response: Thank you for your careful reading of our manuscript. The problem formulation was added in the Introduction section as follows.

When selecting the optimization sand casting process parameters, usually one parameter in batch A reach the ideal value, and another parameter in batch B reach the ideal value, but not all the parameters in one batch can reach the ideal value at the same time. Therefore, it's of great significance to select the optimization process parameters in single batch. The common optimization method of sand casting process parameters is Taguchi's method [12], but this method requires much experimentation and increases costs, which is counter to its purpose of reducing costs. Therefore, it was necessary to find a simple method for optimizing sand casting process parameters. A novel optimization method of process parameters based on gray relational analysis was introduced here.

Once the optimization process parameters were achieved, sand casting should refer to that batch to ensure safe casting and improve casting quality. But if some batches have data missing, they should be deleted and can't be used for optimizing process parameters, which will have a bad influence on optimizing process parameters. What's more, the optimization process parameters are less likely to be achieved due to short of enough valid data. Remeasurement of the missing data needs a lot of money, manpower and time, which would lead to a waste of resources. In addition, it is often difficult to find sand samples from the original casting batch due to the continuous production. Therefore, it was important and necessary here to predict the missing data based on existing data. In the sand casting process, a portion of the recorded parameters data might be lost due to improper care, such as the tally book lost, or the storage data computer breakdown. Here, BP neural network [29] was introduced to predict missing data of sand casting. As there's few papers published refer to the basic principle of BP neural network, therefore correlative theory of BP neural network was deduced to let readers have a better understanding of this model.

Comment 6: There is no range given for the process parameters for optimization.

Response: Thank you for the comments on the paper. We are sorry for not describing the range for the process parameters for optimization. There is interaction effect among these process parameters. With increased compactability, the wet compressive strength of sand increased. After the compactability reached a certain level, the wet compressive strength did not increase much. However, the venting quality decreased rapidly with continued increased compactability. It needs more data support to give the range for the process parameters for optimization. Therefore, we just select the optimization batch

22 based on the given data.

Comment 7: On Page 10 line No 18, GP, GCS "positive indicator" and for MC "negative indicator" on which base, how it is being decided. Line No. 21-23 self contradictory.

Response: Thank you for your careful work. It's our fault that don't explanation clearly in the manuscript. Therefore, we add corresponding content in the paper as follows.

Assuming that all parameters were within a reasonable range, meeting the normal production of sand casting. The VQ and WCS are the larger the better assessment indicators, therefore they are positive indicators. The MC is the smaller the better assessment indicator, so it's negative indicator. There is interaction effect among these process parameters. Com refers to the volume change of green sand under a certain degree of pressure. With increased Com, the WCS of sand increased. After the Com reached a certain level, the WCS did not increase much. But, the VQ decreased rapidly with continued increased Com. If the parameter Com was too large, the parameter VQ would be small; otherwise, if the parameter Com was too small, the parameter WCS would also be small. Therefore, Com belonged to the moderate indicator category, not too large or too small.

Comment 8: Figure No. 4 is not explained.

Response: Thank you for the comments on the paper. We have added the explanation about Figure No. 4 according to you suggestion as follows.

As was shown in Figure 4, the little circle refers to the actual fitting data, the blue line represents the function relationship between target and output of training samples, and the imaginary line represents the output was equal to the target. The function relationship between target and output of training samples was shown in formula 33.

Output $\approx 0.967 * \text{Target} + 0.0033$ (33)

The correlation coefficient of fitting function was $R=0.98037$. The output and target of the training samples were basically equal (Fig. 4). But due to poor correspondence between the training samples, the output and target of individual samples still had great error, which was consistent with the analysis results (Fig. 3).

Comment 9: Model is not formulated.

Response: We are grateful for your suggestion. It's our fault that ignoring this part, therefore we add one section Procedure for model formulated as follows.

Procedure for model formulated

The main procedure for this composite model formulated based on gray relational analysis and BP neural network were as follows.

(1) Determining the weight of evaluation indicators. To avoid the influence of human factors, the weight of evaluation indicators were determined by the objective entropy weight method, which also saved a lot of manpower. The larger variation degree in the evaluation indicators, indicating that the evaluated projects had good discrimination in this regard, and a larger weight should be given to these evaluation indicators.

(2) Selecting the optimization process parameters of sand casting. The GRD can be achieved after the weight of evaluation indicators were determined. The larger the GRD, the closer the evaluated project was to the ideal project. Accordingly, the order of the evaluated projects was confirmed.

(3) Predicting the missing data of process parameters. Dividing the process parameters data into training samples and prediction samples. Using the training samples for training BP neural network, then predicting the missing data based on the trained BP neural network. This paper also provided a new method for determining the number of hidden neurons in a network.

Comment 10: Conclusion is not self explanatory.

Response: Thank you for the comments on this paper. We have revised the Conclusion by logic as follows.

A composite optimization and prediction model of sand casting process parameters was proposed based on gray relational analysis and BP neural network. The main conclusions were as follows.

First, to avoid the influence of human factors, the weights of evaluation indicators were achieved based on the objective entropy weight method and the results were 0.3812, 0.2231, 0.0594, and 0.3364 for the VQ, WCS, MC, and Com, respectively. The weight of parameter VQ was the biggest, indicating that the sand batch had good discrimination in this regard.

Second, the GRDs of foundry sand were obtained according to gray relational analysis after the weight of process parameters were determined, and the results showed that sample batch 22 possessed the largest GRD. Thus, it was advised that process parameters of sand castings refer to batch 22 to ensure safety casting and improve casting quality.

Third, the process parameters data were divided into training samples and prediction samples. The

mean square error of training samples was the smallest when the number of hidden neurons was 9 and the epoch 140 when training BP neural network, and the relative error of the prediction samples was 0.62% based on this trained BP neural network. The relevant theory was deduced before predicting missing data, such that there will be a general understanding regarding the prediction principle of BP neural network.

Fourth, to demonstrate the validity and feasibility of BP neural network adopted in the process of missing data prediction, gray system theory was applied to compare the result of missing data prediction. The relative error of predictive value was 2.64% based on gray system theory. The results showed that the predictive value of BP neural network was more precision than gray system theory.

Comment 11: Some of the references are not referred at all in the text and some are not even relevant to topic itself.

Response: Thank you for your instructive suggestions. We looked through the references carefully to guarantee that all the references were referred in the text, and also we substituted some less relevant references.

Reviewer: 2

Comment 1: There are a lot of issues on the language expression, some sentence patterns are not used correctly.

Response: Thank you for your careful reading of our manuscript. We are very sorry for our language mistakes in the paper. According to the comments from you, we polished the manuscript with a professional assistance in writing.

Comment 2: The relationship between parameters optimization and data missing prediction can't be found. It seems the two key points proposed in this paper are fragmented.

Response: Thank you for your instructive suggestions. We add some transitional content to connect these two parts as follows according to the comments from you.

Once the optimization process parameters were achieved, sand casting should refer to that batch to ensure safe casting and improve casting quality. But if some batches have data missing, they should be deleted and can't be used for optimizing process parameters, which will have a bad influence on optimizing process parameters. What's more, the optimization process parameters are less likely to be achieved due to short of enough valid data. Remeasurement of the missing data needs a lot of money, manpower and time, which would lead to a waste of resources. In addition, it is often difficult to find sand samples from the original casting batch due to the continuous production. Therefore, it was important and necessary here to predict the missing data based on existing data. In the sand casting process, a portion of the recorded parameters data might be lost due to improper care, such as the tally book lost, or the storage data computer breakdown. Here, BP neural network [29] was introduced to predict missing data of sand casting. As there's few papers published refer to the basic principle of BP neural network, therefore correlative theory of BP neural network was deduced to let readers have a better understanding of this model.

Comment 3: Both gray relational analysis and BP neural network are quite conventional methods. This paper devotes much space (about 7 pages) describing the basic principle of common methods, and there is little improvement.

Response: Thank you for the comments on this paper. We really agree with your viewpoints that gray relational analysis and BP neural network are quite conventional methods. In practical application, we usually get the results based on the procedure of model. But if we didn't know the basic principle of model, we could not get the right results. In addition, if we ignored the basic principle of these model in the paper, readers will be confused about how to get results. What's more, there is few papers published in English refer to the basic principle of BP neural network, therefore correlative theory of the BP neural network was deduced to let readers have a better understanding of this model. Also, this paper provided a new method for determining the number of hidden neurons in a network according to the mean square error of training samples.

Comment 4: From my point of view, BP neural network is a simple but not an effective method for missing data prediction in this paper. In fact, people are not interested in the process of network training and its parameter selection. It's meaningless for common methods like BP NN. What is the advantage of the method proposed in this paper? There should be more comparison between different missing data predictions.

Response: Thank you for the comments on this paper. This paper provided a new method for

determining the number of hidden neurons in a network according to the mean square error of training samples, therefore we should have a better understanding of the network training process. The advantage of the method proposed were added in the paper as follows.

The advantages of the composite model proposed in this paper were as follows. First, optimization process parameters can contribute to reducing foundry defects and improving casting quality as well as ensuring safety in sand casting. Second, prediction of missing data can avoid repetitive waste of resources and the difficulties of retesting. Third, this paper provided a new method for determining the number of hidden neurons in a network according to the mean square error of training samples.

The added content of compare with gray system theory was as follow.

4.1 Compare with gray system theory

To demonstrate the validity and feasibility of BP neural network adopted in the process of missing data prediction, this section compare the result of missing data prediction with GM(1,1) model. GM(1,1) model is one basic and important part in the gray system theory, which refers to first order Gray Model in one variable [20]. The brief procedure of GM(1,1) model was as follows [20].

Let the original data sequence be and the accumulation data sequence , where can be calculated based on formula 34.

(34)

Suppose the matrix accord with exponential change law, and the whitenization equation of GM(1,1) model is as follow.

(35)

Let be the initial conditions, solve the equation 35 and the predictive formula of can be achieved, shown in 36.

(36)

The predictive formula of the original data sequence is shown in formula 37, calculated as .

(37)

The parameters a and b are based on the least squares estimate of GM(1,1) model, shown in formula 38.

(38)

where the matrix B and Y are as follow.

The background value is mean sequence of , calculated by formula 39.

(39)

Suppose VQ process parameters of the first 39 batches were the original data sequence, then the predictive formula was shown in formula 40 based on the above procedures.

(40)

Therefore, the VQ predictive value of the 40th batch was 180.1235 according to formula 40, and the relative error was 2.64%.

It can be seen that the predictive value of BP neural network was more precision than gray system theory. It's mainly because that BP neural network takes advantage of more information, and BP neural network is a kind of nonlinear mathematical model that can simulates nonlinear process of any degree. The biggest advantage of this model is that it can be trained and tested repeatedly, and finally it can approach any complicated nonlinear function.

Comment 5: In chapter 4. Discussion, there should be deeper analysis but not similar content like overview.

Response: Thank you for your valuable advice. The Discussion was revised as follows.

4. Discussion

4.1 Compare with gray system theory

To demonstrate the validity and feasibility of BP neural network adopted in the process of missing data prediction, this section compare the result of missing data prediction with GM(1,1) model. GM(1,1) model is one basic and important part in the gray system theory, which refers to first order Gray Model in one variable [20]. The brief procedure of GM(1,1) model was as follows [20].

Let the original data sequence be and the accumulation data sequence , where can be calculated based on formula 34.

(34)

Suppose the matrix accord with exponential change law, and the whitenization equation of GM(1,1) model is as follow.

(35)

Let be the initial conditions, solve the equation 35 and the predictive formula of can be achieved, shown in 36.

(36)

The predictive formula of the original data sequence is shown in formula 37, calculated as .

(37)

The parameters a and b are based on the least squares estimate of GM(1,1) model, shown in formula 38.

(38)

where the matrix B and Y are as follow.

The background value is mean sequence of \bar{x}_k , calculated by formula 39.

(39)

Suppose VQ process parameters of the first 39 batches were the original data sequence, then the predictive formula was shown in formula 40 based on the above procedures.

(40)

Therefore, the VQ predictive value of the 40th batch was 180.1235 according to formula 40, and the relative error was 2.64%.

It can be seen that the predictive value of BP neural network was more precision than gray system theory. It's mainly because that BP neural network takes advantage of more information, and BP neural network is a kind of nonlinear mathematical model that can simulates nonlinear process of any degree. The biggest advantage of this model is that it can be trained and tested repeatedly, and finally it can approach any complicated nonlinear function.

4.2 Discussion of results

The present results confirmed that the composite model proposed in this paper was successfully applied to sand casting. This was the first report in this field of the optimization of process parameters based on gray relational analysis and predictions of missing data based on BP neural network. The advantages of the composite model proposed in this paper were as follows. First, optimization process parameters can contribute to reducing foundry defects and improving casting quality as well as ensuring safety in sand casting. Second, prediction of missing data can avoid repetitive waste of resources and the difficulties of retesting. Third, this paper provided a new method for determining the number of hidden neurons in a network according to the mean square error of training samples.

Motivated by previous studies on gray relational analysis [15–17], this method was introduced into the field of sand casting for process parameters optimizing for the first time in this paper. The larger the GRD, the closer the project evaluated was to the ideal project. Accordingly, the order of the projects evaluated was confirmed. The GRD of sample batch 22 was the largest, so sample batch 22 was the optimization process parameters of sand casting. The results showed that gray relational analysis can be easily used for process parameters optimizing of sand casting. Different from Taguchi's method [12], which requires a lot of experimentations, gray relational analysis is less demanding on the quantity and regularity of samples and can be easily calculated without any discrepancy between calculated results and quantitative analysis results. In the process of determining assessment indicators weights, to eliminate the influence of human factors, such as subjective [19,20] and integrated [24–26] weight methods, the objective entropy weight method [22] was adopted, which also significantly decreased manpower.

In determining the number of hidden neurons, there were not consistent results [38–41]. This paper provided a new method for determining the number of hidden neurons according to the mean square error of training samples. The results showed that mean square error of training samples was the smallest when the number of hidden neurons was 9 and the epoch 140 when training BP neural network, and the relative error of the prediction samples was 0.62%. BP neural network is a widely used prediction technique, but there is few studies refer to the basic principle [27–29], which does not conducive to the improvement of the algorithm. Therefore, relevant theory of BP neural network was deduced before predicting missing data, such that there will be a general understanding of the prediction principle. To demonstrate the validity and feasibility of BP neural network adopted in the process of missing data prediction, gray system theory [20] was applied to compare the result of missing data prediction. The relative error of predictive value was 2.64% based on gray system theory. The results showed that the predictive value of BP neural network was more precision than gray system theory. The composite model proposed in this paper can be used for related research in this field.

To simplify the discussion, the identification coefficient in formula 2 was only set as 0.5. Future study should focus on the influence of the identification coefficient in optimizing process parameters as well as the influence of different training functions in BP neural network on the prediction results.

References

1. Lin B, Wu Y, Zhang L. 2011 Estimates of the potential for energy conservation in the Chinese steel industry. *Energy Policy* 6, 3680–3689. (doi: 10.1016/j.enpol.2011.03.077)
2. Li Y, Chen W, Huang D, Luo J, Liu Z, Chen Y, Liu Q, Su S. 2010 Energy conservation and emissions reduction strategies in foundry industry. *China Foundry* 4, 392–399. (Available at: <https://doaj.org/article/28fe3e4dda7941cfa19700e93c3dbc7f>)

3. Hasanbeigi A, Jiang Z, Price L. 2014 Retrospective and prospective analysis of the trends of energy use in Chinese iron and steel industry. *Journal of Cleaner Production* 74, 105-118. (doi: 10.1016/j.jclepro.2014.03.065)
4. He F, Zhang Q, Lei J, Fu W, Xu X. 2013 Energy efficiency and productivity change of China's iron and steel industry: Accounting for undesirable outputs. *Energy Policy* 54, 204-213. (doi: 10.1016/j.enpol.2012.11.020)
5. Zhang B, Wang ZH, Yin JH, Su LX. 2012 CO₂ emission reduction within Chinese iron & steel industry: practices, determinants and performance. *Journal of Cleaner Production* 33, 167-178. (doi: 10.1016/j.jclepro.2012.04.012)
6. Li G, Ji T. 2016 Severe accidental water vapour explosions in a foundry in China in 2012. *Journal of Loss Prevention in the Process Industries* 41, 55-59. (doi: 10.1016/j.jlp.2016.03.001)
7. Marmo L, Piccinini N, Fiorentini L. 2013 Missing safety measures led to the jet fire and seven deaths at a steel plant in Turin. Dynamics and lessons learned. *Journal of Loss Prevention in the Process Industries* 1, 215-224. (doi: 10.1016/j.jlp.2012.11.003)
8. Porru S, Calza S, Arici C. 2011 An effectiveness evaluation of a multifaceted preventive intervention on occupational injuries in foundries: A 13-year follow-up study with interrupted time series analysis. *Int Arch Occup Environ Health* 8, 867-876. (doi: 10.1007/s00420-011-0638-3)
9. Shi Y, Wang L. 2010 Sand casting precision technology based on non-occupying coating technology. *International Conference On Computer Design and Applications* 2, 156-159. (doi: 10.1109/ICCD.2010.5541174)
10. Tavakoli R, Davami P. 2008 Optimal riser design in sand casting process with evolutionary topology optimization. *Structural and Multidisciplinary Optimization* 2, 205-214. (doi: 10.1007/s00158-008-0282-z)
11. Tavakoli R, Davami P. 2008 Optimal riser design in sand casting process by topology optimization with SIMP method I: Poisson approximation of nonlinear heat transfer equation. *Structural and Multidisciplinary Optimization* 2, 193-202. (doi: 10.1007/s00158-007-0209-0)
12. Kumar S, Satsangi PS, Prajapati DR. 2010 Optimization of green sand casting process parameters of a foundry by using Taguchi's method. *The International Journal of Advanced Manufacturing Technology* 1-4, 23-34. (doi: 10.1007/s00170-010-3029-0)
13. Nekere ML, Singh AP. 2012 Optimization of Aluminium blank sand casting process by using Taguchi's robust design method. *International Journal for Quality Research* 1, 81-97. (Available at: <https://doaj.org/article/f290557d1fa549609d0239ede3c48da1>)
14. Chen WJ, Lin CX, Chen YT, Lin JR. 2016 Optimization design of a gating system for sand casting Aluminium A356 using a Taguchi method and multi-objective culture-based QPSO algorithm. *Advances in Mechanical Engineering* 4, 1-14. (doi: 10.1177/1687814016641293)
15. Huang SJ, Chiu NH, Chen LW. 2008 Integration of the grey relational analysis with genetic algorithm for software effort estimation. *European Journal of Operational Research* 3, 898-909. (doi: 10.1016/j.ejor.2007.07.002)
16. Wei G. 2011 Grey relational analysis model for dynamic hybrid multiple attribute decision making. *Knowledge-Based Systems* 5, 672-679. (doi: 10.1016/j.knosys.2011.02.007)
17. Hashemi SH, Karimi A, Tavama M. 2015 An integrated green supplier selection approach with analytic network process and improved grey relational analysis. *International Journal of Production Economics* 159, 178-191. (doi: 10.1016/j.ijpe.2014.09.027)
18. Kose E, Burmaoglu S, Kabak M. 2013 Grey relational analysis between energy consumption and economic growth. *Grey Systems: Theory and Application* 3, 291-304. (doi: 10.1108/gs-06-2013-0010)
19. Zhang S, Zhu Q. 2014 Heterogeneous wireless network selection algorithm based on group decision. *The Journal of China Universities of Posts and Telecommunications* 3, 1-9. (doi: 10.1016/s1005-8885(14)60294-6)
20. Zangenehmadar Z, Moselhi O. 2016 Prioritizing deterioration factors of water pipelines using Delphi method. *Measurement* 90, 491-499. doi: 10.1016/j.measurement.2016.05.001
21. Aguado R, Arrizabalaga A, Arabiourrutia M, Lopez G, Bilbao J, Olazar M. 2014 Principal component analysis for kinetic scheme proposal in the thermal and catalytic pyrolysis of waste tyres. *Chemical Engineering Science* 106, 9-17. (doi: 10.1016/j.ces.2013.11.024)
22. Delgado A, Romero I. 2016 Environmental conflict analysis using an integrated grey clustering and entropy-weight method: A case study of a mining project in Peru. *Environmental Modelling & Software* 77, 108-121. (doi: 10.1016/j.envsoft.2015.12.011)
23. Gorgij AD, Kisi O, Moghaddam AA, Taghipour A. 2017 Groundwater quality ranking for drinking purposes, using the entropy method and the spatial autocorrelation index. *Environmental Earth Sciences* 7, 1-9. (doi: 10.1007/s12665-017-6589-6)
24. Yu F-C, Fang G-H, Shen R. 2014 Study on comprehensive early warning of drinking water sources for the Gucheng Lake in China. *Environmental Earth Sciences* 9, 3401-3408. (doi: 10.1007/s12665-014-3246-1)

25. Cui Q, Li Y. 2015 An empirical study on energy efficiency improving capacity: The case of fifteen countries. *Energy Efficiency* 6, 1049-1062. (doi: 10.1007/s12053-015-9337-3)
26. Sun R, Wang X, Zhou Z, Ao X, Sun X, Song M. 2014 Study of the comprehensive risk analysis of dam-break flooding based on the numerical simulation of flood routing. Part i: Model development. *Natural Hazards* 3, 1547-1568. (doi: 10.1007/s11069-014-1154-z)
27. Niu B, Liu YJ, Zong GD, Han ZY, Fu J. 2017 Command filter-based adaptive neural tracking controller design for uncertain switched nonlinear output-constrained systems. *IEEE Transactions on Cybernetics* 10, 3160-3171. (doi: 10.1109/TCYB.2016.2647626)
28. Li ZJ, Xia YQ, Su CY, Deng J, Fu J, He W. 2015 Missile guidance law based on robust model predictive control using neural network optimization. *IEEE Transactions on Neural Networks and Learning Systems* 8, 1803-1809. (doi: 10.1109/TNNLS.2014.2345734)
29. Guo ZH, Wu J, Lu HY, Wang JZ. 2011 A case study on a hybrid wind speed forecasting method using BP neural network. *Knowledge-Based Systems* 7, 1048-1056. (doi: 10.1016/j.knosys.2011.04.019)
30. Bezuglov A, Comert G. 2016 Short-term freeway traffic parameter prediction: Application of grey system theory models. *Expert Systems with Applications* 62, 284-292. (doi: 10.1016/j.eswa.2016.06.032)
31. Sun W, Sun J. 2017 Daily PM2.5 concentration prediction based on principal component analysis and LSSVM optimized by cuckoo search algorithm. *J Environ Manage.* 188, 144-152. (doi: 10.1016/j.jenvman.2016.12.011)
32. Noori R, Yeh H-D, Abbasi M, Kachosangi FT, Moazami S. 2015 Uncertainty analysis of support vector machine for online prediction of five-day biochemical oxygen demand. *Journal of Hydrology* 527, 833-843. (doi: 10.1016/j.jhydrol.2015.05.046)
33. Pashah S, Arif AFM. 2014 Fatigue life prediction of adhesive joint in heat sink using Monte Carlo method. *International Journal of Adhesion and Adhesives* 50, 164-175. (doi: 10.1016/j.ijadhadh.2014.01.018)
34. David E. Rumelhart, James L. McClelland and PDP Research Group. 1986 *Parallel Distributed Processing: Exploration in the Microstructures of Cognition*. The MIT Press. (Available at: <https://mitpress.mit.edu/books/parallel-distributed-processing>)
35. David E. Rumelhart, Geoffrey E. Hinton, Ronald J. Williams. 1986 Learning representations by back-propagating errors. *Nature* 323, 533-536. (Available at: <https://www.nature.com/articles/323533a0>)
36. Richard P. Lippmann. 1987 An introduction to computing with neural nets. *IEEE ASSP Magazine* 2, 4-22. (Available at: <http://ieeexplore.ieee.org/document/1165576/>)
37. Chen YJ. 2016 Influence of the gas from cavity on quality of the crankshaft casting during mold filling. Shandong University, Weihai (in Chinese). (Available at: <http://lib.wh.sdu.edu.cn/cn/>)
38. Hahn ME. 2007 Feasibility of estimating isokinetic knee torque using a neural network model. *J Biomech.* 5, 1107-1114. (doi: 10.1016/j.jbiomech.2006.04.014)
39. Onada T. 1995 Neural network information criterion for the optimal number of hidden units. *Proceedings of ICNN'95 - International Conference on Neural Networks 1995*, 275-280. (doi: 10.1109/ICNN.1995.488108)
40. McDonnell MD, Tissera MD, Vladusich T, Van Schaik A, Tapson J. 2015 Fast, simple and accurate handwritten digit classification by training shallow neural network classifiers with the 'extreme learning machine' algorithm. *PLoS One* 8, e0134254. (doi: 10.1371/journal.pone.0134254)
41. Verhagen JV, Scott TR. 2004 Artificial neural network analysis of gustatory responses in the thalamic taste relay of the rat. *Physiology & Behavior* 4, 499-513. (doi: 10.1016/j.physbeh.2003.10.005)

Appendix B

Dear Editor and Reviewers,

We thank you for giving us constructive suggestions which will help us to improve the quality of the paper. Here we submit a new version of our manuscript, which has been modified according to the your instructive suggestions.

The point to point responds to the your comments are listed as following

Comments to Authors:

Associate Editor Comments to Author (Professor Jun Fu):

A reviewer pointed out that this manuscript lacked the information about the application of entropy to the available data and the clarification for the process of optimization. Additionally, sentences in this paper seem to have been borrowed from other published articles according to the result of iThenticate for plagiarism checking. Thus my recommendation is Reject & allow resubmission.

Reviewer comments to Author:

Reviewer: 3

Comments to the Author(s)

the English need revisions, and the process of optimization needs clarification, there is no information about the application of entropy to the available data

Response: Thank you for your instructive suggestion. The manuscript has been revised according to your advice as follows.

(1) there is no information about the application of entropy to the available data

Response: Thank you for your valuable advice. The information about the application of entropy to the available data has been added in the manuscript as follows.

In order to get the GRD of evaluation indicators, their weights should be know first. The weights of the evaluation indicators, including VQ, WCS, MC, and Com, were achieved by entropy weight method based on formulas (4)–(6), shown in Table 2.

Table 2. Weights of the evaluation indicators (using original data of evaluation indicators in Table 1)

Indicators	VQ	WCS	MC	Com
e_j	0.9496	0.9505	0.9501	0.9498
w_j	0.252	0.2475	0.2495	0.251

(2) the process of optimization needs clarification,

Response: Thank you for your valuable and thoughtful comment. The process of optimization has been added in the manuscript as follows.

3.1. Optimization of performance parameters based on gray relational analysis

Sand casting performance parameters have a significant impact on casting quality, such that selection of the appropriate performance parameters was needed. Taking 40 batches of floor sand

in the casting line, the original data of performance parameters in this foundry sand sample were measured, including venting quality (VQ), wet compressive strength (WCS), moisture content (MC), and compactability (Com) (Table 1) [37]. Among these performance parameters, VQ is the ability that gas to penetrate high molecular material under a certain degree of pressure and time; WCS refers to the ability of object to resist external pressure under the saturated water condition; MC is the percentage of water content to the total mass of the object; Com is the proportion of the volume change of the object under a certain degree of pressure.

Table 1. Original data and gray relational coefficient of foundry sand

Batch No.	Original data				Gray relational coefficient			
	VQ ($\text{m}^4 \cdot \text{N}^{-1} \cdot \text{S}^{-1}$)	WCS (MPa)	MC (%)	Com (%)	VQ	WCS	MC	Com
1	170	0.155	3.69	40	0.4286	0.7218	0.674	0.705
2	175	0.16	3.56	38	0.5	1	0.9394	0.5036
3	160	0.15	3.65	40	0.3333	0.5647	0.7382	0.705
4	175	0.155	3.63	40	0.5	0.7218	0.7751	0.705
5	170	0.155	3.6	40.5	0.4286	0.7218	0.8379	0.7833
6	165	0.16	3.66	42	0.375	1	0.721	0.8704
7	170	0.15	3.68	41	0.4286	0.5647	0.689	0.8812
8	180	0.155	3.6	41.5	0.6	0.7218	0.8379	0.993
9	170	0.16	3.72	43	0.4286	1	0.6327	0.698
10	180	0.15	3.7	43	0.6	0.5647	0.6596	0.698
11	175	0.16	3.67	42	0.5	1	0.7046	0.8704
12	185	0.15	3.61	42.5	0.75	0.5647	0.8158	0.7747
13	175	0.155	3.64	41	0.5	0.7218	0.7562	0.8812
14	180	0.155	3.67	43	0.6	0.7218	0.7046	0.698
15	170	0.155	3.71	42	0.4286	0.7218	0.6459	0.8704
16	180	0.15	3.64	42	0.75	0.5647	0.8379	0.8704
17	175	0.15	3.59	41.5	0.5	0.5647	0.8611	0.993
18	180	0.15	3.54	41	0.6	0.5647	1	0.8812
19	175	0.155	3.64	42	0.5	0.7218	0.7562	0.8704
20	180	0.155	3.62	42	0.6	0.7218	0.7949	0.8704
21	175	0.15	3.6	38	0.5	0.5647	0.8379	0.5036
22	190	0.155	3.66	41.5	1	0.7218	0.721	0.993
23	170	0.15	3.66	39.5	0.4286	0.5647	0.721	0.6409
24	185	0.15	3.63	41	0.75	0.5647	0.7751	0.8812
25	180	0.16	3.72	43	0.6	1	0.6327	0.698
26	180	0.15	3.62	40.5	0.6	0.5647	0.7949	0.7833
27	170	0.155	3.7	41.5	0.4286	0.7218	0.6596	0.993
28	175	0.15	3.66	42.5	0.5	0.5647	0.721	0.7747
29	180	0.15	3.58	40	0.6	0.5647	0.8857	0.705
30	180	0.15	3.74	44	0.6	0.5647	0.6079	0.5826
31	180	0.15	3.67	43	0.6	0.5647	0.7046	0.698
32	185	0.145	3.62	41	0.75	0.4637	0.7949	0.8812
33	180	0.15	3.7	42	0.6	0.5647	0.6596	0.8704

34	180	0.15	3.67	42	0.6	0.5647	0.7046	0.8704
35	175	0.155	3.62	41.5	0.5	0.7218	0.7949	0.993
36	175	0.15	3.6	41.5	0.5	0.5647	0.8379	0.993
37	175	0.155	3.72	42.5	0.5	0.7218	0.6327	0.7747
38	175	0.145	3.72	43	0.5	0.4637	0.6327	0.698
39	175	0.15	3.64	41	0.5	0.5647	0.7562	0.8812
40	185	0.15	3.6	42	0.6	0.5647	0.7562	0.8704

In Table 1, assuming that all parameters were within a reasonable range, meeting the normal production of sand casting. The VQ and WCS are the larger the better assessment indicators, therefore they are positive indicators. The MC is the smaller the better assessment indicator, so it's negative indicator. There is interaction effect among these performance parameters. Com refers to the volume change of green sand under a certain degree of pressure. With increased Com, the WCS of sand increased. After the Com reached a certain level, the WCS did not increase much. But, the VQ decreased rapidly with continued increased Com. If the parameter Com was too large, the parameter VQ would be small; otherwise, if the parameter Com was too small, the parameter WCS would also be small. Therefore, Com belonged to the moderate indicator category, not too large or too small.

The original data matrix was nondimensionalized according to formula (1), and the ideal project $D^* = [1.0765 \ 1.0483 \ 0.9702 \ 1]$ was achieved, where the optimal value of Com was the average of the nondimensionalized data.

The gray relational coefficient of evaluation indicators can be obtained based on formula (2), shown in Table 1.

In order to get the GRD of evaluation indicators, their weights should be know first. The weights of the evaluation indicators, including VQ, WCS, MC, and Com, were achieved by entropy weight method based on formulas (4)–(6), shown in Table 2.

Table 2. Weights of the evaluation indicators (using original data of evaluation indicators in Table 1)

Indicators	VQ	WCS	MC	Com
e_j	0.9496	0.9505	0.9501	0.9498
w_j	0.252	0.2475	0.2495	0.251

The GRD of foundry sand was achieved based on formulas (3) and the weights of the evaluation indicators (Table 3).

Table 3. GRD of foundry sand

Batch No.	GRD	Batch No.	GRD	Batch No.	GRD	Batch No.	GRD
1	0.6318	11	0.7678	21	0.6012	31	0.642
2	0.7343	12	0.7268	22	0.8598	32	0.7233
3	0.5849	13	0.7145	23	0.5885	33	0.674
4	0.675	14	0.6808	24	0.7433	34	0.6852
5	0.6923	15	0.6663	25	0.7318	35	0.7522
6	0.7404	16	0.7563	26	0.6859	36	0.7241
7	0.6409	17	0.7299	27	0.7005	37	0.657
8	0.7881	18	0.7616	28	0.6401	38	0.5738
9	0.6886	19	0.7118	29	0.6889	39	0.6756
10	0.6307	20	0.7466	30	0.5889	40	0.6981

As shown in Table 2, the GRD of sample batch 22 was the largest, which was the closest to the

ideal project. Thus, it was advised that the performance parameters of sand casting refer to batch 22 to ensure safe casting and improve casting quality.

(3) the English need revisions

Response: Thank you for your careful reading of our manuscript. The grammatical errors in the manuscript have been revised.

(4) Additionally, sentences in this paper seem to have been borrowed from other published articles according to the result of iThenticate for plagiarism checking.

Response: Thank you for your comment. The repetitive contents in the manuscript have been revised.

Appendix C

Dear Editor and Reviewers,

We thank you for giving us constructive suggestions which will help us to improve the quality of the paper. Here we submit a new version of our manuscript, which has been modified according to your instructive suggestions.

The point to point responds to the your comments are listed as following

Reviewer: 4

Comments to the Author(s)

Comment 1: Clarity of equations and figures (Like Figure 1, 3, 4) need to be improved.

Response: Thank you for your careful reading of our manuscript, and sorry for our improper written in the paper. All the equations in the manuscript have been improved. But after I submit the revised paper to the system, part of the equations in the PDF version were still messy code (the capital letters L and M are ellipsis). If this manuscript can be accepted for publication, the publishing editor will handle these issues.

In addition, all the figures in the manuscript have been improved, and the resolution ratio of these original figures are 600 DPI * 600 DPI, shown as follows. But after I submit the revised paper to the system, the resolution ratio of the figures in the PDF version were transformed into 300 DPI * 300 DPI.

Figure 1. Structural chart of BP neural network.

Figure 2. Relationship between mean square error and number of hidden neurons.

Figure 3. Relationship between mean square error and epoch.

Figure 4. Regression analysis of training samples.

Comment 2: Explanation is required for each equation to improve the readability of the paper.

Response: Thank you for your instructive suggestion. Explanation is added for each equation to improve the readability of the paper. Following each equation, corresponding explanations have been added (highlighted in blue in Main Document (tracked changes) .doc).

Comment 3: Original data set need to be added as Appendix

Response: Thank you for your valuable advice. The original data of foundry sand have been added as Appendix A (Table S1), shown as follows.

Appendix A

Table S1. Original data of foundry sand

Batch No.	performance parameters			
	VQ (m ⁴ ·N ⁻¹ ·S ⁻¹)	WCS (MPa)	MC (%)	Com (%)
1	170	0.155	3.69	40
2	175	0.16	3.56	38
3	160	0.15	3.65	40
4	175	0.155	3.63	40
5	170	0.155	3.6	40.5
6	165	0.16	3.66	42
7	170	0.15	3.68	41
8	180	0.155	3.6	41.5

9	170	0.16	3.72	43
10	180	0.15	3.7	43
11	175	0.16	3.67	42
12	185	0.15	3.61	42.5
13	175	0.155	3.64	41
14	180	0.155	3.67	43
15	170	0.155	3.71	42
16	180	0.15	3.64	42
17	175	0.15	3.59	41.5
18	180	0.15	3.54	41
19	175	0.155	3.64	42
20	180	0.155	3.62	42
21	175	0.15	3.6	38
22	190	0.155	3.66	41.5
23	170	0.15	3.66	39.5
24	185	0.15	3.63	41
25	180	0.16	3.72	43
26	180	0.15	3.62	40.5
27	170	0.155	3.7	41.5
28	175	0.15	3.66	42.5
29	180	0.15	3.58	40
30	180	0.15	3.74	44
31	180	0.15	3.67	43
32	185	0.145	3.62	41
33	180	0.15	3.7	42
34	180	0.15	3.67	42
35	175	0.155	3.62	41.5
36	175	0.15	3.6	41.5
37	175	0.155	3.72	42.5
38	175	0.145	3.72	43
39	175	0.15	3.64	41
40	185	0.15	3.6	42

Reviewer: 5

Comments to the Author(s)

Comment 1: Give more importance for the experiment and not on the basis of the optimization tool

Response: Thank you for your valuable and thoughtful comment. More importance for the experiment have been added in the manuscript as follows.

Among these performance parameters, VQ is the ability that gas to penetrate high molecular material under a certain degree of pressure and time. The venting capacity of sand casting is not only increased by the riser and gas vent, but also by the VQ of the foundry sand. The VQ of foundry sand should not be too low, so as to avoid the occurrence of boiling and pore defects in

the casting process. But, the VQ of foundry sand should not be too high, so as to prevent the molten metal from infiltrating into the porosity, which will cause rough surface or abreuvage of casting. Therefore, the VQ of the foundry sand needs to be within an appropriate range, and should not be too high or too low. For high density molding, the VQ of foundry sand should be high. For low and medium density molding, the VQ of foundry sand should be low.

WCS refers to the ability of object to resist external pressure under the saturated water condition. If the WCS of foundry sand is insufficient, the sand mould may be damaged or collapsed during the process of drawing and mould assembling; in the pouring process, the sand mould may not withstand the impact of molten metal, which will cause blisters or even molten metal discharging from the parting surface. However, the WCS of foundry sand is not the higher the better. The higher WCS of foundry sand needs more bentonite, which not only affects the MC and VQ of foundry sand, but also increases the cost of casting. In addition, the higher WCS of foundry sand brings difficulties to the process of sand milling and shakeout.

MC is the percentage of water content to the total mass of the object. If the MC of foundry sand is low, the VQ of foundry sand will be high, and the casting is prone to sand burning. If the MC of foundry sand is high, a large amount of gas will be generated in the cavity due to evaporation of moisture during the pouring process. Once the gas in the cavity cannot be discharged smoothly within a limited time, an explosion accident may occur. Therefore, the MC of foundry sand should have a suitable range according to the filed practice.

Com is the proportion of the volume change of the object under a certain degree of pressure. On the one hand, the Com of foundry sand should not be too small, otherwise the bentonite will be insufficiently wetted, leading the foundry sand to brittleness, low surface strength and difficulty in drawing. On the other hand, the Com of foundry sand should not be too large, otherwise the castings are prone to boiling and pore defects.

In Table S1, assuming that all the performance parameters were within a reasonable range, meeting the normal production of sand casting. From the above analysis, it is know that all the performance parameters, namely VQ, WCS, MC and Com, belonged to the moderate indicator category, not too large or too small.

Comment 2: More justification needed on why this optimization tool is used and why not others.

Response: Thank you for your valuable advice. More justification have been added in the manuscript on why this optimization tool is used and why not others, shown as follows.

When optimization the sand casting performance parameters, usually one parameter in batch *A* reach the ideal value, and another parameter in batch *B* reach the ideal value, but not all the parameters in one batch can reach the ideal values at the same time. Therefore, it's of great significance to optimization the performance parameters in single batch. The common optimization method of sand casting performance parameters is Taguchi's method [14], but this method has some disadvantages. First, the test number of orthogonal array in Taguchi's method is too much, which requires much experimentation and increases costs, and this is counter to its purpose of reducing costs. Second, the purpose of Taguchi's method is to reduce the effects of mutagenic factors rather than removing the mutagenic factors to improve quality. Third, despite the large amount of data, we are still unable to obtain any information about the interaction between controllable variable factors. Fourth, if there is no interaction between the controllable factors and interference factors, then a sound design does not exist. Therefore, it's necessary to

find a simple method for optimizing sand casting performance parameters. A novel optimization method of performance parameters based on gray relational analysis was introduced here.

Gray relational analysis is a very active branch in gray system theory. Its basic idea is to determine the gray relational degree between different sequences according to the geometrical shape of the sequence curves [15]. The larger the gray relational degree, the closer the project evaluated was to the ideal project. Accordingly, the order of the projects evaluated can be confirmed. Gray relational analysis does not require too many samples, nor does the sample have a typical distribution law, and the workload of calculation is relatively small. The gray relational analysis results are in good agreement with the qualitative analysis results. Gray relational analysis has been applied to many fields, such as decision-making [16], green supplier selection [17], and quality evaluation of red wine [18]. Wei [16] has investigated the dynamic hybrid multiple attribute decision-making problems based on gray relational analysis. In this study, gray relational analysis was used for optimization of sand casting performance parameters. During the process of gray relational analysis, assessment indicator weights should first be calculated.

Comment 3: Why authors have chosen the factors of performance parameters and what all other possible performance parameters in sand casting explain in details.

Response: Thank you for your instructive suggestion. All other possible performance parameters in sand casting have been added in the revision paper, and explanation has been added on why we chose these four factors of performance parameters, shown as follows.

3.1. Selection of Sand casting performance parameters

The foundry sand has many performance parameters, and each of them has an impact on the casting quality. However, the influence of each performance parameter on the casting quality is not the same, and the testing frequency is smaller for the performance parameter which is more important for improving the casting quality. The sand casting performance parameters and their testing frequency are shown in Table 1.

Table 1. Sand casting performance parameters and their testing frequency

performance parameters	sampling spot	testing frequency
venting quality (VQ), wet compressive strength (WCS), moisture content (MC), compactability (Com)	discharge port of sand mill, or conveyer of foundry sand	once every half to two hours
	under the hopper of molding machine	once every four to five hours
content of effective braize, content of effective bentonite, wet-heat tensile strength	under the hopper of molding machine	once a day
content of clay, content of lump, grain composition	under the hopper of molding machine	once a week
sand temperature, availability of bentonite, mobility, fracture and heat shock time	under the hopper of molding machine	in case of need

As is shown in Table 1, the sand casting performance parameters such as VQ, WCS, MC and Com are the most important factors for improving the casting quality, and the testing frequency is also the smallest. Therefore, this study optimizes the performance parameters such as VQ, WCS, MC and Com for improving casting quality.